# Memory-Efficient Gradient Unrolling for Large-Scale Bi-level Optimization

Qianli Shen[1][*]    Yezhen Wang[1]    Zhouhao Yang[1]    Xiang Li[1]    Haonan Wang[1]
Yang Zhang[1]    Jonathan Scarlett[1]    Zhanxing Zhu[2]    Kenji Kawaguchi[1]
[1]National University of Singapore    [2]University of Southampton, UK

## Abstract

Bi-level optimization (BO) has become a fundamental mathematical framework for addressing hierarchical machine learning problems. As deep learning models continue to grow in size, the demand for scalable bi-level optimization has become increasingly critical. Traditional gradient-based bi-level optimization algorithms, due to their inherent characteristics, are ill-suited to meet the demands of large-scale applications. In this paper, we introduce **F**orward **G**radient **U**nrolling with **F**orward **G**radient, abbreviated as $(\mathbf{FG})^2\mathbf{U}$, which achieves an unbiased stochastic approximation of the meta gradient for bi-level optimization. $(FG)^2U$ circumvents the memory and approximation issues associated with classical bi-level optimization approaches, and delivers significantly more accurate gradient estimates than existing large-scale bi-level optimization approaches. Additionally, $(FG)^2U$ is inherently designed to support parallel computing, enabling it to effectively leverage large-scale distributed computing systems to achieve significant computational efficiency. In practice, $(FG)^2U$ and other methods can be strategically placed at different stages of the training process to achieve a more cost-effective two-phase paradigm. Further, $(FG)^2U$ is easy to implement within popular deep learning frameworks, and can be conveniently adapted to address more challenging black-box bi-level optimization scenarios. We provide a thorough convergence analysis and a comprehensive practical discussion for $(FG)^2U$, complemented by extensive empirical evaluations, showcasing its superior performance in diverse large-scale bi-level optimization tasks. Code is available at `https://github.com/ShenQianli/FG2U`.

## 1 Introduction

Bi-level optimization is a mathematical framework with a long history of research [10, 65, 73], dealing with hierarchical optimization problems where one problem is nested within the other. A bi-level optimization problem can be formulated as:

$$\min_{\boldsymbol{\phi}} \ f(\boldsymbol{\theta}^*(\boldsymbol{\phi}), \boldsymbol{\phi}) \quad s.t. \ \boldsymbol{\theta}^*(\boldsymbol{\phi}) \in \arg\min_{\boldsymbol{\theta}} g(\boldsymbol{\theta}, \boldsymbol{\phi}), \tag{1}$$

where $\boldsymbol{\theta} \in \Theta \subseteq \mathbb{R}^M$ denotes the inner parameter, $\boldsymbol{\phi} \in \Phi \subseteq \mathbb{R}^N$ denotes the meta parameter, and $f$, $g$ are called the meta objective function and inner objective function, respectively.

Recently, with the rise of deep learning, bi-level optimization has regained attention as a theoretical framework covering a wide range of machine learning problems, including hyperparameter optimization [46, 43, 17, 16, 45], neural architecture search [78, 38, 14], robust machine learning [79, 76, 71, 26], meta learning [15, 53, 49, 2], and physics-informed machine learning [23, 62]. In these scenarios, the inner problem often pertains to the optimization of neural networks, thereby

---

[*]Correspondence: `shenqianli@u.nus.edu`

38th Conference on Neural Information Processing Systems (NeurIPS 2024).

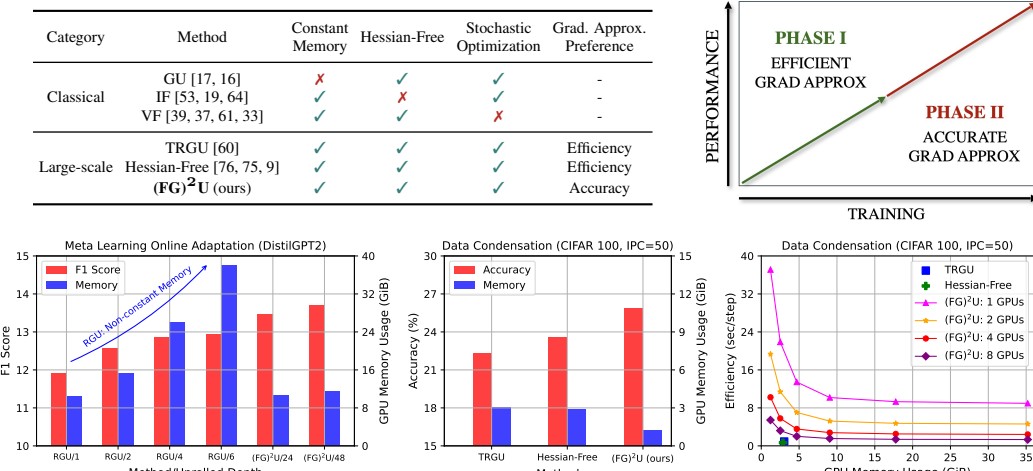

Figure 1: **Top Left**: A comparison of bi-level optimization methods. $(FG)^2U$ circumvents the large-scale challenges inherent in classical bi-level optimization techniques. Within large-scale bi-level optimization, $(FG)^2U$ prioritizes the accuracy of gradient approximation over efficiency. **Top Right**: An overview of the cost-effective two-phase paradigm. $(FG)^2U$ is ideally positioned in Phase II to enhance performance after an approximate solution has been obtained using other efficient methods. **Bottom Left**: GPU Memory Usage and Performance on *Meta Learning Online Adaptation* experiment. $(FG)^2U$ can effectively address the memory issue of RGU when both the inner model and the unrolled depth are large. **Bottom Center**: GPU Memory Usage and Performance on *Data Condensation* experiments. The performance of $(FG)^2U$ surpasses that of other large-scale bi-level optimization methods, owing to its accurate gradient approximation, while demonstrating better memory efficiency. **Bottom Right**: Efficiency tradeoff of $(FG)^2U$ on *Data Condensation* experiments. The efficiency of $(FG)^2U$ can be well enhanced via intra/inter-GPU parallelism.

precipitating challenges associated with gradient-based bi-level optimization. Consequently, various gradient-based bi-level optimization algorithms have been developed [73]. These algorithms typically employ an iterative solution $\theta_T$ obtained by executing multiple inner optimization steps to approximate the meta gradient, and provide different tradeoffs between computational costs and performance for meta gradient approximation.

However, as the scale of deep learning models continues to expand, the requirements for scalability in bi-level optimization correspondingly increase. Existing gradient-based bi-level optimization algorithms, due to their inherent characteristics, are ill-suited to meet the demands of large-scale applications. Concretely, *gradient unrolling* (GU) methods [17, 16, 40, 60] are bottlenecked by the memory overhead associated with either the dimension of the inner parameter or the number of iterative steps for the inner problem. Implicit Function (IF) approaches [48, 19, 64, 76] are compromised by approximation errors, which stem from the iterative estimation of inner solutions and computations that involve the Hessian matrix. *Value Function* (VF) based strategies [39, 37, 61, 33], although exhibit commendable theoretical properties [8] for deterministic bi-level optimization, have yet to gain traction in practical applications, predominantly due to their limitations in addressing large-scale stochastic challenges [73]. Recent advancements in algorithms [60, 9] have been specifically tailored for large-scale bi-level optimization. Although these methodologies facilitate efficient gradient approximation by compromising accuracy, they may result in significantly suboptimal performance due to biased gradient approximations. Additionally, these methods struggle in more complex scenarios, such as when inner problems are addressed through black-box optimization.

In this paper, we propose a novel method called **F**orward **G**radient **U**nrolling with **F**orward **G**radient, abbreviated as $(FG)^2U$, which achieves an unbiased stochastic approximation of the meta gradient for bi-level optimization. $(FG)^2U$ circumvents the memory issues associated with GU-based approaches and approximation issues associated with IF-based approaches. Compared to recently developed large-scale bi-level optimization approaches, $(FG)^2U$ delivers significantly more accurate gradient estimates. Additionally, $(FG)^2U$ is inherently designed to support parallel computing, enabling it to effectively leverage large-scale distributed computing systems to achieve significant computational efficiency. In practice, a cost-effective two-phase paradigm can be achieved by strategically placing

$(FG)^2U$ and other methods at different stages of the training process to balance efficiency and performance. Further, $(FG)^2U$ is easy to implement within popular deep learning frameworks, and can be conveniently adapted to address more challenging zeroth-order bi-level optimization scenarios.

We provide an overview of $(FG)^2U$ in Figure 1 to illustrate its strengths and role in large-scale bi-level optimization. The rest of the paper is organized as follows. Firstly, in Section 2, we provide summaries of existing bi-level optimization algorithms and discuss their limitations in large-scale contexts. Next, in Section 3, we introduce the proposed method, $(FG)^2U$, followed by a convergence analysis in Section 3.1 and a detailed discussion of the practical considerations in Section 3.2. Further, in Section 4, we conduct extensive empirical studies covering large-scale bi-level optimization in computer vision, natural language processing, and physics-informed machine learning to demonstrate the efficacy of $(FG)^2U$ in large-scale bi-level optimization scenarios.

## 2 Background

**Gradient-based Bi-level Optimization.** Within deep learning applications, the model concerned with optimizing over $\boldsymbol{\theta}$ as presented in (1) typically constitutes deep neural networks. The optimal parameters of such networks are not explicitly accessible and are estimated through iterative procedures. Consequently, the primal problem of bi-level optimization in (1) is approximately reformulated as follows:

$$\min_{\boldsymbol{\phi} \in \Phi} \; h(\boldsymbol{\phi}) := f(\boldsymbol{\theta}_T(\boldsymbol{\phi}), \boldsymbol{\phi}), \tag{2}$$

$$\text{where } \boldsymbol{\theta}_0(\boldsymbol{\phi}) = \boldsymbol{\Omega}_0(\boldsymbol{\phi}), \; \boldsymbol{\theta}_t(\boldsymbol{\phi}) = \boldsymbol{\Omega}_t(\boldsymbol{\theta}_{t-1}(\boldsymbol{\phi}), \boldsymbol{\phi}) \in \Theta, \; t = 1, \dots, T,$$

where $\Phi \subseteq \mathbb{R}^N$, $\Theta \subseteq \mathbb{R}^M$ are the parameter spaces; $T$, commonly called the unrolled depth, denotes the number of inner optimization steps for approximating $\boldsymbol{\theta}^*(\boldsymbol{\phi})$; $\boldsymbol{\Omega}_0 : \mathbb{R}^N \to \mathbb{R}^M$ specifies the initialization of the inner optimization, and $\boldsymbol{\Omega}_t : \Theta \times \Phi \to \Phi$ delineates the transition dynamics of the inner optimization at timestep $t$. In particular, for gradient descent, $\boldsymbol{\Omega}_t(\boldsymbol{\theta}_{t-1}(\boldsymbol{\phi}), \boldsymbol{\phi}) = \boldsymbol{\theta}_{t-1} - \eta_t \nabla_{\boldsymbol{\theta}} g(\boldsymbol{\theta}_{t-1}, \boldsymbol{\phi})$, where $\eta_t$ denotes the step size at timestep $t$.

To optimize $\boldsymbol{\phi}$ using a first-order method, it is necessary to estimate the meta gradient $\nabla_{\boldsymbol{\phi}} h$, which can be further decomposed according to the chain rule:

$$\underbrace{\nabla_{\boldsymbol{\phi}} h(\boldsymbol{\phi})}_{\text{meta gradient}} = \underbrace{\frac{\partial f(\boldsymbol{\theta}_T(\boldsymbol{\phi}), \boldsymbol{\phi})}{\partial \boldsymbol{\theta}_T} \frac{d\boldsymbol{\theta}_T(\boldsymbol{\phi})}{d\boldsymbol{\phi}}}_{\text{implicit gradient}} + \underbrace{\frac{\partial f(\boldsymbol{\theta}_T(\boldsymbol{\phi}), \boldsymbol{\phi})}{\partial \boldsymbol{\phi}}}_{\text{explicit gradient}}. \tag{3}$$

The computation of meta-gradient poses a significant challenge, primarily due to the need for efficient approximation of the implicit gradient. This task is complicated by the recursive dependency of $\boldsymbol{\theta}_T$ on $\boldsymbol{\phi}$. To surmount this challenge, a variety of gradient-based bi-level optimization algorithms have been developed, as extensively reviewed recently in [73]. These algorithms can be fundamentally categorized into three types based on their approach to meta-gradient approximation: *Gradient Unrolling* (GU), *Implicit Function* (IF), and *Value Function* (VF). Recent innovations such as truncated RGU (TRGU) [60] and Hessian-Free approaches [76, 75, 9], which are predicated on GU and IF methodologies respectively, have introduced significant biases in their approximations to accommodate the computational constraints of large-scale scenarios. In the subsequent paragraph, we furnish a concise overview of GU-based approaches, addressing their non-constant memory issues in large-scale applications. Extended discussions on the remaining methods are reserved for Appendix B.

**Gradient Unrolling.** The core idea behind GU [17, 16, 40, 60] entails unrolling the inner optimization into an expansive computational graph, followed by the employment of automatic differentiation (AD) techniques for the iterative computation of gradients.

*Forward Gradient Unrolling* (FGU) [17, 16] computes the meta gradient using the following forward recursive formula, starting from $\boldsymbol{Z}_0 = \frac{d\boldsymbol{\Omega}_0(\boldsymbol{\phi})}{d\boldsymbol{\phi}}$:

$$\underbrace{\frac{d\boldsymbol{\theta}_t(\boldsymbol{\phi})}{d\boldsymbol{\phi}}}_{\boldsymbol{Z}_t} = \underbrace{\frac{\partial \boldsymbol{\Omega}_t(\boldsymbol{\theta}_{t-1}(\boldsymbol{\phi}), \boldsymbol{\phi})}{\partial \boldsymbol{\theta}_{t-1}}}_{\boldsymbol{A}_t} \underbrace{\frac{d\boldsymbol{\theta}_{t-1}(\boldsymbol{\phi})}{d\boldsymbol{\phi}}}_{\boldsymbol{Z}_{t-1}} + \underbrace{\frac{\partial \boldsymbol{\Omega}_t(\boldsymbol{\theta}_{t-1}(\boldsymbol{\phi}), \boldsymbol{\phi})}{\partial \boldsymbol{\phi}}}_{\boldsymbol{B}_t}, \; t = 1, \dots, T, \tag{4}$$

*Reverse Gradient Unrolling* (RGU) [46, 16], instead of the employment of explict reccursive formulas of $\boldsymbol{Z}_T$, focuses on the implicit reccursive formulas of $\nabla_{\boldsymbol{\phi}} h$:

$$
\nabla_{\boldsymbol{\phi}} h(\boldsymbol{\phi}) = \underbrace{\frac{\partial f(\boldsymbol{\theta}_T(\boldsymbol{\phi}), \boldsymbol{\phi})}{\partial \boldsymbol{\theta}_T}}_{\boldsymbol{d}_T} \underbrace{\frac{d\boldsymbol{\theta}_T(\boldsymbol{\phi})}{d\boldsymbol{\phi}}}_{\boldsymbol{Z}_T} + \underbrace{\frac{\partial f(\boldsymbol{\theta}_T(\boldsymbol{\phi}), \boldsymbol{\phi})}{\partial \boldsymbol{\phi}}}_{\boldsymbol{c}_T}
$$

$$
= \boldsymbol{d}_T \boldsymbol{Z}_T + \boldsymbol{c}_T \overset{(4)}{=} \underbrace{\boldsymbol{d}_T \boldsymbol{A}_T}_{\boldsymbol{d}_{T-1}} \boldsymbol{Z}_{T-1} + \underbrace{\boldsymbol{d}_T \boldsymbol{B}_T + \boldsymbol{c}_T}_{\boldsymbol{c}_{T-1}} = \cdots = \boldsymbol{d}_0 \boldsymbol{Z}_0 + \boldsymbol{c}_0. \tag{5}
$$

The corresponding reverse recursive formulas can thus be summarized as

$$
\boldsymbol{c}_{t-1} = \boldsymbol{c}_t + \boldsymbol{d}_t \boldsymbol{B}_t, \quad \boldsymbol{d}_{t-1} = \boldsymbol{d}_t \boldsymbol{A}_t, \quad t = T, \ldots, 1. \tag{6}
$$

**Weakness (GU): Non-Constant Memory**. Both GU approaches exhibit a non-constant memory overhead, which constrains their utility in large-scale scenarios. The forward reccursive formulas in (4) revolve around the Jacobian matrix product, demanding $\mathcal{O}(MN)$ space consumption. The reverse recursive formulas in (6) necessitate the storage of the entire trajectory of the inner optimization $\boldsymbol{\theta}_{0:T}$ for backward computation, thereby imposing a memory requirement of $\mathcal{O}(TM)$. These requirements are often impractical for large-scale bi-level optimization, when $\boldsymbol{\phi}$ and $\boldsymbol{\theta}$ are of high dimension and a significant unrolled depth is required.

**Forward Gradient.** Forward-mode automatic differentiation (forward-mode AD) has been applied to a variety of research fields, including the training of recurrent neural networks [70], the computation of Hessian vector products [50], etc. However, the computation of the true gradient via forward-mode AD requires the full Jacobian, which is typically too costly to compute.

To solve this, forward gradient learning [69, 4, 63, 4, 56], built upon forward-mode AD, was proposed. Forward gradient methods update parameters based on the directional gradient along a random perturbation direction for backpropagation-free training. More formally, given a differentiable function $h : \mathbb{R}^N \to \mathbb{R}$, the gradient for a given input $\boldsymbol{\phi} \in \mathbb{R}^N$ can be approximated as

$$
\hat{\nabla} h(\boldsymbol{\phi}) = \nabla h(\boldsymbol{\phi}) \boldsymbol{v} \boldsymbol{v}^T, \tag{7}
$$

where $\boldsymbol{v} \sim p(\boldsymbol{v})$ is a $N$-dimensional multivariate random variable, satisfying $\mathbb{E}[\boldsymbol{v}\boldsymbol{v}^T] = \mathbf{I}$. Common choices of the distribution of $\boldsymbol{v}$ include Rademacher $\boldsymbol{v} \sim \mathrm{Unif}(\{-1, 1\}^N)$, Gaussian $\boldsymbol{v} \sim \mathcal{N}(\mathbf{0}, \boldsymbol{I})$, and uniform distribution over a set of normalized orthogonal coordinates $\boldsymbol{v} \sim \mathrm{Unif}(\{\sqrt{N}\boldsymbol{e}_i\}_{1:N})$. For any given $\boldsymbol{\phi}$, $\hat{\nabla} h(\boldsymbol{\phi})$ is an unbiased estimator of $\nabla h(\boldsymbol{\phi})$, as $\mathbb{E}[\hat{\nabla} h(\boldsymbol{\phi})] = \mathbb{E}[\nabla h(\boldsymbol{\phi}) \boldsymbol{v}\boldsymbol{v}^T] = \nabla h(\boldsymbol{\phi})\mathbb{E}[\boldsymbol{v}\boldsymbol{v}^T] = \nabla h(\boldsymbol{\phi})\mathbf{I} = \nabla h(\boldsymbol{\phi})$. Despite the unbiasedness of $\hat{\nabla} h$, the dimension-dependent variance of the estimated gradient with a single direction impedes the scaling-up to high-dimensional problems. In practice, Monte Carlo gradient estimation can be used via averaged forward gradients over multiple random directions to reduce the variance.

# 3   $(FG)^2U$: Forward Gradient Unrolling with Forward Gradient

We aim to circumvent the memory overhead issues associated with forward gradient unrolling (FGU) as discussed in Section 2. We begin by examining the forward gradient of $h$ at $\boldsymbol{\phi}$,

$$
\hat{\nabla} h(\boldsymbol{\phi}) = \nabla h(\boldsymbol{\phi}) \boldsymbol{v} \boldsymbol{v}^T \overset{(5)}{=} (\boldsymbol{d}_T \boldsymbol{Z}_T \boldsymbol{v} + \boldsymbol{c}_T \boldsymbol{v}) \boldsymbol{v}^T, \tag{8}
$$

where $\boldsymbol{v} \sim p(\boldsymbol{v})$ is a $N$-dimensional multivariate random variable, satisfying $\mathbb{E}[\boldsymbol{v}\boldsymbol{v}^T] = \mathbf{I}$. We follow the idea of FGU introduced in Section 2 to compute $\boldsymbol{Z}_T \boldsymbol{v}$. By multiplying both sides of (4) by $\boldsymbol{v}$ on the right, we can obtain the recursive formulas for $\boldsymbol{Z}_t \boldsymbol{v}$ as

$$
\boldsymbol{Z}_0 \boldsymbol{v} = \boldsymbol{B}_0 \boldsymbol{v}; \quad \boldsymbol{Z}_t \boldsymbol{v} = \boldsymbol{A}_t \boldsymbol{Z}_{t-1} \boldsymbol{v} + \boldsymbol{B}_t \boldsymbol{v}, \quad t = 1, \ldots, T. \tag{9}
$$

The revised recursive formulas in (9) facilitate the tracking of a $M$-dimensional vector $\boldsymbol{Z}_t \boldsymbol{v}$, rather than full Jacobian $\boldsymbol{Z}_t$ of size $M \times N$, throughout the forward pass. The stochastic estimation in (8) is unbiased, adhering to the properties of forward gradient methods. To reduce the variance, we use

Monte Carlo estimate via averaged forward gradients over $b$ *i.i.d.* random directions:

$$\hat{\nabla}h(\phi) = \frac{1}{b}\sum_{i=1}^{b}\nabla h(\phi)\boldsymbol{v}_i\boldsymbol{v}_i^T = \frac{1}{b}\sum_{i=1}^{b}(\boldsymbol{d}_T\boldsymbol{Z}_T\boldsymbol{v}_i + \boldsymbol{c}_T\boldsymbol{v}_i)\boldsymbol{v}_i^T. \tag{10}$$

We call this algorithm **(FG)$^2$U**, as an abbreviation of **F**orward **G**radient **U**nrolling with **F**orward **G**radient. The algorithm is summarized in Appendix A as Algorithm 1.

Compared to GU-based methods, as discussed in Section 2, (FG)$^2$U eliminates the dependency on the meta parameter dimension $N$ and the depth of unrolling $T$ without introducing bias, significantly enhancing memory efficiency. Unlike IF-based methods, as discussed in Appendix B.2, (FG)$^2$U overcomes the approximation issues associated with them while maintaining a constant memory overhead, thus providing superior gradient approximation. Compared to TRGU and Hessian-Free methods, which compromise approximation accuracy for efficiency, (FG)$^2$U consistently delivers accurate gradient approximations. The computational efficiency of (FG)$^2$U can be further enhanced by leveraging large-scale distributed computing resources, capitalizing on its inherently parallelizable formulation as presented in (10). In practice, a more cost-effective two-phase paradigm can be achieved by strategically placing (FG)$^2$U and other methods at different stages of the training process, as we will discuss in Section 3.2. For an illustration of the role of (FG)$^2$U in large-scale bi-level optimization, please refer to Figure 1.

## 3.1 Convergence

In this section, we provide a convergence analysis for (FG)$^2$U. The proofs can be found in Appendix C.

First, we establish a bound on the variance of the estimated gradient, when employing random vectors whose entries follow the Rademacher distribution.

**Lemma 3.1.** *For any $\phi \in \Phi$, if $\boldsymbol{v}_i \sim \mathrm{Unif}(\{-1, 1\}^N)$, the gradient estimation in (10), satisfies*

$$\mathbb{E}\|\hat{\nabla}h(\phi) - \nabla h(\phi)\|^2 = \frac{1}{\rho}\|\nabla h(\phi)\|^2,$$

*where $\rho := \frac{b}{N-1} \in (0, 1]$ as the sample size $b$ is selected from $1, \cdots, N-1$.*

The resultant error is bounded by $O\left(\frac{N-1}{b}\right)$, where $b$ represents the sample size used for computing the forward gradient, and $N$ is the dimensionality of the gradient itself. This bound demonstrates how the error scales inversely with the sample size while also being influenced by the gradient's dimensionality.

Next, we lay down the following assumptions, on which our main theorems are based. Let $\psi = (\boldsymbol{\theta}, \phi) \in \Theta \times \Phi$ denote the combination of the lower-level parameter $\boldsymbol{\theta}$ and the meta parameter $\phi$. Following existing papers on the theory of bilevel optimization [45, 60, 28], in Assumption 3.2, we adopt some standard assumptions over the smoothness of the objective functions $f$ and $g$.

**Assumption 3.2.** *The meta objective function $f(\psi)$ and the lower-level objective function $g(\psi)$ are both $C$-Lipschitz and $L$-smooth, i.e., for any $\psi, \psi' \in \Theta \times \Phi$,*

$$|f(\psi) - f(\psi')| \leq C\|\psi - \psi'\|, \quad \|\nabla f(\psi) - \nabla f(\psi')\| \leq L\|\psi - \psi'\|, \tag{11}$$

$$|g(\psi) - g(\psi')| \leq C\|\psi - \psi'\|, \quad \|\nabla g(\psi) - \nabla g(\psi')\| \leq L\|\psi - \psi'\|. \tag{12}$$

The next assumption regulates that the transition functions $\boldsymbol{\Omega}$ satisfy similar smoothness conditions.

**Assumption 3.3.** *The transition functions $\boldsymbol{\Omega}_{0:T}$ are $C_{\boldsymbol{\Omega}}$-Lipschitz and $L_{\boldsymbol{\Omega}}$-smooth, i.e., for any $\phi, \phi' \in \Phi$,*

$$\|\boldsymbol{\Omega}_0(\phi) - \boldsymbol{\Omega}_0(\phi')\| \leq C_{\boldsymbol{\Omega}}\|\phi - \phi'\|, \ \|\nabla\boldsymbol{\Omega}_0(\phi) - \nabla\boldsymbol{\Omega}_0(\phi')\| \leq L_{\boldsymbol{\Omega}}\|\phi - \phi'\|. \tag{13}$$

*For any $\psi, \psi' \in \Theta \times \Phi$, $t = 1, \ldots, T$,*

$$\|\boldsymbol{\Omega}_t(\psi) - \boldsymbol{\Omega}_t(\psi')\| \leq C_{\boldsymbol{\Omega}}\|\psi - \psi'\|, \ \|\nabla\boldsymbol{\Omega}_t(\psi) - \nabla\boldsymbol{\Omega}_t(\psi')\| \leq L_{\boldsymbol{\Omega}}\|\psi - \psi'\|. \tag{14}$$

Assumption 3.3 is made to ensure the generality of our analysis over different optimizers. Note that $\boldsymbol{\Omega}$ is scheme-dependent w.r.t. the gradient-based optimizer we adopt for lower-level problems. In many cases, such as gradient descent where $\boldsymbol{\Omega}_t(\psi_{t-1}) = \boldsymbol{\theta}_{t-1} - \eta_t \nabla_{\boldsymbol{\theta}} g(\psi_{t-1})$, Assumption 3.3 is a direct consequence of Assumption 3.2.

We propose the following theorem and remark for convergence analysis of (FG)²U on problem (2). Notice the convergence result can be extended to the primal BO problem (1) with some further assumptions. We place a proof scratch and some discussions in Appendix C.3.

**Theorem 3.4** (Convergence). *Suppose that Asumption 3.2 and Assumption 3.3 hold. Setting the learning rate $\beta = \frac{\rho}{(\rho+1)L_h}$ for gradient descent over the hyperparameter $\phi$, then there exists a constant $L_h$ (depending on $C$, $L$, $C_\Omega$, $L_\Omega$, and $T$, and defined formally in the proof) such that*

$$\frac{1}{K} \sum_{k=0}^{K-1} \mathbb{E}\left[\|\nabla h(\phi_k)\|^2\right] \leq \frac{4L_h\left(\mathbb{E}[h(\phi_0)] - \min_\phi h(\phi)\right)}{\rho K}. \tag{15}$$

**Remark 3.5.** *Theorem 3.4 shows that Algorithm 1 converges to an $\epsilon$-accurate stationary point with a convergence rate of $= \mathcal{O}(\epsilon^{-1}\rho^{-1})$.*

Recall that $\rho = \frac{b}{N-1}$, which indicates that the convergence rate is linearly dependent on $N$, which poses a significant challenge when managing high-dimensional meta-parameters $\phi$. However, it is important to note that the dimension-dependent convergence rate represents an upper bound, and scalability has been found to be feasible with several practical considerations, as discussed in the following subsection.

## 3.2 Practical Considerations

**Choice of** $b$. According to the convergence analysis in Section 3.1, a sample size of $b = \mathcal{O}(N)$ is required to achieve a convergence rate of $\mathcal{O}(\epsilon^{-1})$. However, it has been widely observed that forward gradient and zeroth-order optimization, despite having dimension-dependent convergence rates, work well empirically with $b = \mathcal{O}(1)$ in large-scale scenarios, such as in LLM fine-tuning [47, 74]. In this paper, we select the largest possible $b$ that does not exceed the GPU memory limit for our empirical study. Additionally, gradient accumulation is utilized to further control variance and stabilize the training process.

**Cost-Effective Two-Phase Paradigm**. It is important to note that the upper bound delineated in (15) linearly depends on the performance discrepancy between the initialized meta parameter $\phi_0$ and the optimal. This dependence motivates the adoption of a more cost-effective two-phase paradigm for large-scale bi-level optimization. In the initial phase, we utilize efficient yet less accurate gradient approximation methods, such as TRGU [60] and Hessian-Free [9], to efficiently establish an initial $\phi_0$ that surpasses random initialization, while keeping computational overhead manageable. Subsequently, in the second phase, (FG)²U is utilized for a more accurate, albeit less efficient, gradient approximation to further elevate the performance, leveraging extensive computational resources.

**Implementation**. The technique employed in computing $\nabla h(\phi)v$ is identified as forward-mode automatic differentiation (forward-mode AD). In advanced automatic differentiation libraries, such as JAX [5] and PyTorch [3], forward-mode AD is efficiently implemented as Jacobian-vector product (jvp), without the necessity of explicitly computing the Jacobian matrix. The FLOP cost of jvp is approximately three times that of a standard forward pass, while the memory overhead is doubled. In practice, it is only necessary to define the forward computational graph of inner optimization and invoke forward-mode AD, which simplifies the implementation process significantly. Regarding distributed training, JAX offers the vmap interface for efficient intra-GPU parallelism and the pmap interface for effective inter-GPU parallelism.

**Zeroth-order Bi-level optimization**. In certain applications of bi-level optimization, the inner problem is approached as a black box, where the gradient of $\Omega$ is inaccessible, rendering the analytical gradient unrolling unfeasible. For example, in PDE-constrained optimization [23, 62], in which the inner problem entails solving a Partial Differential Equation (PDE) using a non-differentiable solver. In such scenarios, rather than employing forward-mode Automatic Differentiation (AD), one can resort to Finite Difference methods to approximate the directional gradient $\nabla h(\phi)v$ by

$$\nabla h(\phi)v = \lim_{\mu \to 0} \frac{h(\phi + \mu v) - h(\phi)}{\mu} \approx \frac{h(\phi + \bar{\mu}v) - h(\phi)}{\bar{\mu}} \tag{16}$$

with sufficiently small positive $\bar{\mu} > 0$. We refer to this zeroth-order variant of (FG)²U as (FG)²U-ZO, noting that the computation solely encompasses two forward passes and does not involve the utilization of any first-order information. The memory complexity is the same as forward-mode AD

and the actual computation time will be slightly less than forward-mode AD, at the cost of introducing an approximation bias. We give a more detailed discussion within the context of *zeroth-order optimization* [41] in Appendix D, and empirically study a corresponding case in Section 4.

## 4  Experiments

We conduct experiments across various contexts, as detailed in the respective subsections. Initially, we engage in an image data condensation task, where we focus on a comprehensive performance comparison between $(FG)^2U$ and both classical and large-scale bi-level optimization algorithms. Subsequently, we investigate meta-learning for the online adaptation of language models, employing a GPT model as the inner model, to illustrate how $(FG)^2U$ effectively circumvents the non-constant memory issue associated with RGU. Finally, we address a physics-informed bi-level optimization problem, where gradient-based inner solvers are ineffective, to demonstrate the efficacy of combining $(FG)^2U$-ZO, the zeroth-order variant of $(FG)^2U$ discussed in Section 3.2, with non-differentiable numerical solvers.

**Data Condensation**.  To overcome the challenges posed by large-scale datasets, a line of works known as data condensation [68, 72] has been proposed. The main idea is to generate a compact, synthesized dataset, designed to elicit similar behaviors in machine learning models as those trained with the original, massive dataset. The objective of the mainstream principles [72] designed for data condensation can be naturally formulated as a bi-level optimization problem. We focus on the best-known principle *performance matching* [72] on classification tasks, which can be formulated as

$$\min_{\mathcal{D}_c} \mathcal{L}(\theta_T; \mathcal{D}_o), \quad \text{where } \theta_t = \theta_{t-1} - \eta \nabla \mathcal{L}(\theta_{t-1}; \mathcal{D}_c), \ t = 1, \ldots, T, \tag{17}$$

where $\mathcal{D}_o, \mathcal{D}_c$ respectively denote the original and condensed dataset, $\theta$ denotes the model parameter, $\mathcal{L}$ denotes the cross-entropy loss function, and $\eta$ represents the step-size for inner optimization.

| Dataset | IPC | Ratio (%) | Approaches | | | | For Reference | |
|---------|-----|-----------|------|-------------|---------|----------|-----|-------|
| | | | TRGU | Hessian-Free | Neumann | $(FG)^2U$ | RGU | WHOLE |
| MNIST | 1 | 0.017 | $73.76_{\pm 1.68}$ | $65.98_{\pm 1.38}$ | $68.37_{\pm 1.44}$ | $\mathbf{82.44}_{\pm 0.68}$ | $92.32_{\pm 0.33}$ | |
| | 10 | 0.17 | $94.05_{\pm 0.33}$ | $94.97_{\pm 0.34}$ | $95.75_{\pm 0.24}$ | $\mathbf{96.12}_{\pm 0.28}$ | $96.79_{\pm 0.29}$ | $99.6_{\pm 0.00}$ |
| | 50 | 0.83 | $96.63_{\pm 0.41}$ | $96.34_{\pm 0.31}$ | $96.78_{\pm 0.22}$ | $\mathbf{97.01}_{\pm 0.19}$ | $97.72_{\pm 0.23}$ | |
| CIFAR-10 | 1 | 0.02 | $20.78_{\pm 1.07}$ | $19.72_{\pm 1.28}$ | $21.33_{\pm 0.90}$ | $\mathbf{29.37}_{\pm 0.75}$ | $34.08_{\pm 0.55}$ | |
| | 10 | 0.2 | $44.01_{\pm 0.57}$ | $45.32_{\pm 1.02}$ | $47.67_{\pm 0.87}$ | $\mathbf{50.10}_{\pm 0.56}$ | $53.15_{\pm 0.53}$ | $84.8_{\pm 0.10}$ |
| | 50 | 1 | $49.22_{\pm 0.45}$ | $48.73_{\pm 0.78}$ | $50.02_{\pm 0.69}$ | $\mathbf{51.98}_{\pm 0.44}$ | $56.37_{\pm 0.37}$ | |
| CIFAR-100 | 1 | 0.2 | $3.96_{\pm 0.68}$ | $3.14_{\pm 0.41}$ | $4.52_{\pm 0.56}$ | $\mathbf{8.22}_{\pm 0.45}$ | $15.61_{\pm 0.32}$ | |
| | 10 | 2 | $20.20_{\pm 0.66}$ | $19.01_{\pm 0.84}$ | $20.87_{\pm 0.82}$ | $\mathbf{23.38}_{\pm 0.33}$ | $25.42_{\pm 0.45}$ | $56.2_{\pm 0.30}$ |
| | 50 | 10 | $22.33_{\pm 0.93}$ | $23.59_{\pm 0.71}$ | $24.52_{\pm 0.77}$ | $\mathbf{25.84}_{\pm 0.31}$ | $28.52_{\pm 0.53}$ | |

Table 1: The performance (testing accuracy %) comparison among various bilevel optimization methods on the data condensation task over three datasets. All the datasets are condensed using a 3-layer ConvNet. IPC: image(s) per class. Ratio (%): the ratio of condensed examples to the whole training set.

We conducted our experiments following the standard data condensation setting established by [68, 77, 67]. A more detailed task description is given in Appendix E.1 and implementation details are given in Appendix F.1.

The condensed datasets are evaluated using 3-layer convolutional networks with randomly initialized parameters, and the average accuracies on test datasets are summarized in Table 1. Compared to large-scale bi-level optimization methods like TRGU and Hessian-Free, which prioritize efficiency at the expense of approximation accuracy, $(FG)^2U$ exhibits significantly better performance, due to more accurate gradient approximation as explained in Appendix B. Additionally, we assessed Neumann Series (denoted as Neumann in Table 1), an IF-based method that mitigates gradient approximation errors through extended computations, as introduced in Appendix B.2. While it demonstrates performance enhancements over the Hessian-Free method, Neumann still yields suboptimal performance

compared to $(FG)^2U$, owing to the inherent bias of the IF-based method. Further discussions and supporting evidence are available in Appendix B.2.

The results of RGU, which represent the upper performance bound for both TRGU and $(FG)^2U$, are provided for reference, along with the results from training on the entire dataset (denoted as WHOLE in Table 1), representing the upper performance bound for all approaches. However, it is crucial to acknowledge that RGU is not practical in large-scale bi-level optimization scenarios due to its non-constant memory requirements, as discussed in Section 2. This limitation will be further exemplified in the subsequent, where the inner model is significantly larger. In principle, the performance of $(FG)^2U$ can be further improved to approach that of RGU by increasing the number of random directions for gradient approximation.

The memory and computational efficiencies of TRGU, Hessian-Free, and $(FG)^2U$ in the most challenging case (CIFAR-100, IPC=50) are reported in Figure 1 (Bottom Right), demonstrating that the efficiency of $(FG)^2U$ can be significantly enhanced through intra/inter-GPU parallelism.

**Meta Learning Online Adaptation of Language Models**. The online adaptation of language models (LM) has been studied recently to keep the knowledge of LM current [34, 27]. However, trivial auto-regressive fine-tuning the LM, which applies uniform weights to all tokens, often results in suboptimal performance in downstream tasks. This issue stems from the default average negative log-likelihood (NLL) loss, which fails to capture the significance of tokens [25]. To overcome this limitation, [25] proposed Context-aware Meta-learned Loss Scaling (CaMeLS), a strategy that employs meta-learning to adjust token weights for more effective online adaptation. Specifically, they meta train a weight model to reweight the auto-regressive loss during online fine-tuning, aiming to enhance LM performance on downstream question-answering tasks. A comprehensive task description and the mathematical formulation of the objectives are detailed in Appendix E.2.

The trained weight model is subsequently fine-tuned on unseen online documents and evaluated on corresponding question-answering tasks. In [25], RGU is utilized for meta gradient approximation. To mitigate the non-constant memory issue associated with RGU, a DistilGPT2 model [59] is chosen as the surrogate base model for training the weight model, instead of larger models typically employed for online adaptation. Additionally, a very limited unrolled depth of 6 is utilized within a 40 GiB GPU memory budget. In our experiments, since $(FG)^2U$ has circumvented the non-constant memory issue associated with RGU, we are able to increase the unrolled depth and upscale the base model for training the weight model. Empirical evaluations are conducted on two datasets, StreamingQA [36] and SQuAD-Seq [54].

| Model (# params) | Method | StreamingQA | | SQuAD-Seq | |
|---|---|---|---|---|---|
| | | EM (↑) | F1 (↑) | EM (↑) | F1 (↑) |
| DistilGPT2 (82M) | CaMeLS + RGU [25, 66] | 1.62 | 5.79 | 1.45 | 3.08 |
| | CaMeLS + RGU (impl.) | 2.04 | 5.53 | 1.52 | 3.16 |
| | CaMeLS + **$(FG)^2U$** (ours) | **2.22** | **6.37** | **1.72** | **3.50** |
| GPT2-Large (774M) | CaMeLS + RGU [25, 66] | 5.35 | 10.60 | 4.97 | 8.63 |
| | CaMeLS + RGU (impl.) | 7.02 | 12.19 | 4.86 | 8.57 |
| | CaMeLS + **$(FG)^2U$** (ours) | **7.21** | **12.50** | **5.56** | **8.99** |
| GPT2-XL (1.5B) | CaMeLS + RGU [25, 66] | 6.55 | 11.67 | 6.70 | 10.15 |
| | CaMeLS + RGU (impl.) | 7.93 | 12.94 | 6.71 | 9.65 |
| | CaMeLS + **$(FG)^2U$** (ours) | **8.89** | **14.42** | **7.37** | **10.37** |

Table 2: Comparison of the online adaptation performance. The reported evaluation metrics include the exact match (EM) and F1 scores. For vanilla CaMeLS [25], RGU is conducted with unrolled depth 6, using DistilGPT2 as the base model. We present both the results reported by [66] and those from our implementation (denoted as impl.). For CaMeLS + $(FG)^2U$, we select unrolled depths from $\{24, 48\}$, and the base model from {DistilGPT2, GPT2}. We report the results for the combination that yields the best F1 score. Additional details and ablation studies are documented in Appendix G.1.

Firstly, we increased the unrolled depth while maintaining the base model as a DistilGPT2. We plotted the F1 scores and GPU memory usages for RGU with unrolled depths of $\{1, 2, 4, 6\}$ and $(FG)^2U$ with unrolled depths of $\{24, 48\}$ on StreamingQA in Figure 1 (Bottom Left). The performance of the weight model is positively correlated with the unrolled depth, substantiating the benefits of

training with larger unrolled depths. The non-constant memory issue associated with RGU can be observed when the unrolled depth increases, while $(FG)^2U$ maintains constant memory even with large unrolled depth. Subsequently, we endeavored to upscale the base model to GPT2 to reduce the disparity between training and evaluation. The performances are summarized in Table 2, with detailed ablation studies on unrolled depths and base model variants documented in Table G.1 and Table G.2.

**Data-driven Discovery of Partial Differential Equations (PDEs).** Let us consider the following general forms of parametrized and nonlinear PDEs:

$$u_t + \mathcal{N}[u; \boldsymbol{\phi}] = 0, \ x \in \Psi, t \in [0, T], \tag{18}$$

where $x$ denotes the space-time coordinate, $\Psi$ denotes a bounded domain with boundary, $u : [0, T] \times \Psi \to \mathbb{R}$ denotes the latent solution, $u_t$ represents the first-order derivative of $u$ with respect to $t$, and $\mathcal{N}$ is a general differential operator parameterized by $\boldsymbol{\phi}$, acting on $\Psi$. This setup encompasses a broad spectrum of problems in physics. For example, the one-dimensional Burgers' equation is defined by $\mathcal{N}[u; \boldsymbol{\phi}] = \mu u u_x - \nu u_{xx}$, where $\boldsymbol{\phi} = (\mu, \nu) \in \mathbb{R}^2$, and $u_x$, $u_{xx}$ represent the first and second-order derivatives of $u$ with respect to $x$, respectively.

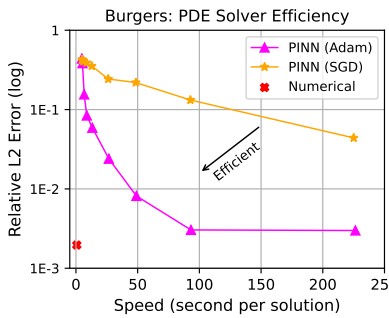

| Method | | $(FG)^2U$ | $(FG)^2U$-ZO |
|---|---|---|---|
| Inner solver | | PINN [52] | Numerical |
| **Burgers** | $\epsilon_\phi(\text{E-2},\downarrow)$ | $2143.58_{\pm 855.26}$ | $\mathbf{0.97}_{\pm 0.45}$ |
| | $\epsilon_u(\text{E-3},\downarrow)$ | $336.06_{\pm 46.91}$ | $\mathbf{0.63}_{\pm 0.33}$ |
| **Allen-Cahn** | $\epsilon_\phi(\text{E-2},\downarrow)$ | $438.13_{\pm 101.77}$ | $\mathbf{2.34}_{\pm 0.64}$ |
| | $\epsilon_u(\text{E-3},\downarrow)$ | $133.61_{\pm 35.93}$ | $\mathbf{0.97}_{\pm 0.54}$ |
| **KdV** | $\epsilon_\phi(\text{E-2},\downarrow)$ | $94.40_{\pm 4.31}$ | $\mathbf{0.72}_{\pm 0.57}$ |
| | $\epsilon_u(\text{E-3},\downarrow)$ | $832.81_{\pm 67.01}$ | $\mathbf{2.72}_{\pm 1.55}$ |

Figure 2: **Left**: Comparison of efficiency between the PINN solver and the numerical solver. We evaluated Adam [29] and SGD as the inner optimizers for the PINN solver, with steps ranging from 100 to 50,000. The results demonstrate that the numerical solver is significantly more efficient. **Right**: Comparison of relative L2 errors in the prediction of $\phi$ and $u$. $\epsilon_\phi = \|\phi_{pred} - \phi\|_2 / \|\phi\|_2$, $\epsilon_u = \|u_{pred} - u\|_2 / \|u\|_2$.

The problem of data-driven discovery of PDEs [52] can be framed as follows: given a set of scattered observations of the latent solution $u(x)$, what are the parameters most accurately describing the observed data? The problem can be formulated as a PDE-constrained optimization problem (PDECO):

$$\min_{\boldsymbol{\phi}} \ \mathbb{E}_{x,u \sim \mathcal{D}} |u(x; \boldsymbol{\phi}) - u|^2 \quad s.t. \quad u_t + \mathcal{N}[u(\cdot; \boldsymbol{\phi}); \boldsymbol{\phi}] = 0, x \in \Psi, \tag{19}$$

where $\mathcal{D} = \{(x_i, u_i)\}_{1:k}$ denotes the observed data. In cases where the closed-form solutions of the nonlinear PDEs are intractable, parametric solutions $u_{\boldsymbol{\theta}}$ are used to approximate the latent solution $u$ for given $\boldsymbol{\phi}$. The PDECO in (19) is then reformulated into a bi-level optimization problem:

$$\min_{\boldsymbol{\phi}} \ \mathbb{E}_{x,u \sim \mathcal{D}} |u_{\boldsymbol{\theta}_S(\boldsymbol{\phi})}(x; \boldsymbol{\phi}) - u|^2 \quad s.t. \quad \boldsymbol{\theta}_s(\boldsymbol{\phi}) = \boldsymbol{\Omega}_s(\boldsymbol{\theta}_{s-1}, \boldsymbol{\phi}), s = 1, \dots, S. \tag{20}$$

Employing gradient-based PDE solvers, such as physics-informed neural networks (PINN) [52], facilitates the direct application of $(FG)^2U$. However, as demonstrated in Figure 2 (Left), the accuracy and efficiency of PINNs fall short of the rigorous demands of scientific computing. This limitation has prompted us to integrate faster and more accurate traditional solvers like the spectral method [1] (see also Appendix E.3.4) to tackle the inner problem. Given these solvers are non-differentiable, we employ $(FG)^2U$-ZO, the zeroth-order variant of $(FG)^2U$ introduced in Section 3.2, to solve the problem.

We conduct experiments on three non-linear PDEs: Burgers, Allen-Cahn, and KdV, with a more detailed task description available in Appendix E.3. The results are summarized in Figure 2 (Right). We can observe that the combination of $(FG)^2U$-ZO and the numerical solver significantly outperforms $(FG)^2U$ and the PINN solver, in terms of both the prediction on $\phi$ and $u$. The implementation details are documented in Appendix F.3.

# 5    Conclusion

In this work, we propose a novel algorithm **Forward Gradient Unrolling with Forward Gradient**, abbreviated as **(FG)$^2$U**, designed to tackle the challenges associated with large-scale bi-level optimization. We conduct a convergence analysis of (FG)$^2$U, perform extensive comparisons with existing methods, and provide detailed discussions on its practical applications. Additionally, we undertake an empirical evaluation across a series of large-scale bi-level optimization tasks. Our findings indicate that (FG)$^2$U effectively complements existing bi-level optimization algorithms, addressing gaps in large-scale bi-level optimization scenarios.

**Limitations and future works**. The experiments conducted in this paper are of relatively small scale, with the largest inner model being a GPT-2 model. We look forward to validating its effectiveness on larger-scale bi-level optimization tasks. Additionally, the application of black-box bi-level optimization and the potential of (FG)$^2$U-ZO remain underexplored, considering the prevalent black-box interaction between users and models today. We hope our work will inspire further development of large-scale bi-level optimization algorithms and their application in corresponding scenarios. Furthermore, we have not specifically addressed the efficiency issues inherited by (FG)$^2$U from the forward gradient method. Enhancing the efficiency of (FG)$^2$U while maintaining its gradient estimation accuracy will be an important direction for future research.

# 6    Acknowledgements

This research is supported by the National Research Foundation Singapore under the AI Singapore Programme (AISG Award No: AISG2-TC-2023-010-SGIL) and the Singapore Ministry of Education Academic Research Fund Tier 1 (Award No: T1 251RES2207, T1 251RES2218). The computational work for this article was partially performed on resources of the National Supercomputing Centre, Singapore (https://www.nscc.sg).

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

# A    Algorithm

---

**Algorithm 1 (FG)$^2$U: Forward Gradient Unrolling with Forward Gradient**

---

**Require:** Initial inner parameters $\boldsymbol{\theta}_0$, initial meta parameter $\boldsymbol{\phi}_0$, random direction distribution $\boldsymbol{p}$,
number of random directions $b$, total meta steps $K$, meta update mappings $\boldsymbol{\Psi}_{1:K}$.

1: $\boldsymbol{\theta} \leftarrow \boldsymbol{\theta}_0, \boldsymbol{\phi} \leftarrow \boldsymbol{\phi}_0$
2: **for** $k = 1, \ldots, K$ **do**
3:     **for** $i = 1, \ldots, b$ **do**
4:         Sample $\boldsymbol{v}_i \sim \boldsymbol{p}(\cdot)$ and initialize $\boldsymbol{y}_i \leftarrow \frac{\partial \boldsymbol{\Omega}_0(\boldsymbol{\theta}, \boldsymbol{\phi})}{\partial \boldsymbol{\phi}} \boldsymbol{v}_i$
5:     **end for**
6:     **for** $t = 1, \ldots, T$ **do**
7:         $\boldsymbol{\theta} \leftarrow \boldsymbol{\Omega}_t(\boldsymbol{\theta}, \boldsymbol{\phi}), \boldsymbol{A} \leftarrow \frac{\partial \boldsymbol{\Omega}_t(\boldsymbol{\theta}, \boldsymbol{\phi})}{\partial \boldsymbol{\theta}}, \boldsymbol{B} \leftarrow \frac{\partial \boldsymbol{\Omega}_t(\boldsymbol{\theta}, \boldsymbol{\phi})}{\partial \boldsymbol{\phi}}$
8:         **for** $i = 1, \ldots, b$ **do**
9:             $\boldsymbol{y}_i \leftarrow \boldsymbol{A}\boldsymbol{y}_i + \boldsymbol{B}\boldsymbol{v}_i$
10:         **end for**
11:     **end for**
12:     **for** $i = 1, \ldots, b$ **do**
13:         $w_i \leftarrow \frac{\partial f(\boldsymbol{\theta}, \boldsymbol{\phi})}{\partial \boldsymbol{\theta}} \boldsymbol{y}_i + \frac{\partial f(\boldsymbol{\theta}, \boldsymbol{\phi})}{\partial \boldsymbol{\phi}} \boldsymbol{v}_i$
14:     **end for**
15:     $\boldsymbol{\phi} \leftarrow \boldsymbol{\Psi}_k(\boldsymbol{\phi}, \frac{1}{b} \sum_{i=1}^b w_i \boldsymbol{v}_i^T)$
16: **end for**
17: **return** $\boldsymbol{\phi}$

---

# B    Extended Discussion on Bi-level Optimization

## B.1    Truncated Reverse Gradient Unrolling (TRGU)

To address the memory issue of GU methods, truncated Reverse Gradient Unrolling (TRGU) [60] is proposed to reduce the memory usage by preserving only the last $K$ steps of the inner optimization trajectory. However, this introduces a significant bias in large-scale scenarios, particularly when the permissible $K$ is small.

Recall (5) and (6), where the conventional RGU method computes the hypergradient by fully unrolling the $T$-step inner optimization into a computational graph. Instead, TRGU performs $s$-step truncated back-propagation and approximates the gradient with the intermediate term $\boldsymbol{c}_{T-s}$:

$$\boldsymbol{c}_{T-s} = \boldsymbol{c}_T + \sum_{t=T-s+1}^{T} \boldsymbol{B}_t \boldsymbol{A}_{t+1} \cdots \boldsymbol{A}_T d_T. \tag{21}$$

According to Proposition 3.1 in [60], if the inner-level objective function $g$ is $L$-smooth, twice-differentiable and globally $\alpha$-strongly convex, and the gradient update rule writes $\boldsymbol{\theta}_t = \boldsymbol{\theta}_{t-1} - \eta \nabla_{\boldsymbol{\theta}} g(\boldsymbol{\theta}_{t-1}, \boldsymbol{\phi})$, then the bias of $s$-step TRGU would be bounded by

$$\|\nabla_{\boldsymbol{\phi}} h - \boldsymbol{c}_{T-s}\| \leq \frac{(1 - \eta\alpha)^s}{\eta\alpha} \|\boldsymbol{d}_T\| \max_{t \in 0, \ldots, T-s} \|\boldsymbol{B}_t\|. \tag{22}$$

The bound (22) demonstrates an exponentially decaying rate in $s$ over the bias of $s$-step TRGU. However, when $s$ gets smaller, which means that we truncate the computational graph heavier in pursuit of lower memory cost, the bias would grow exponentially. This would result in an inaccurate calculation of the hypergradient. Contrastively, our **(FG)$^2$U** is an unbiased estimator of the hypergradient, while still keeping high memory efficiency with a small sample size of forward gradient as in (10).

## B.2    Implicit Function (IF)

Another idea for computing the implicit gradient is to utilize the implicit function theorem (IFT) [30]. Suppose that the inner optimality $\nabla_{\boldsymbol{\theta}} g(\boldsymbol{\theta}_T(\boldsymbol{\phi}), \boldsymbol{\phi}) \approx 0$ is approximately achieved by sufficient inner optimization steps. If $g$ is second-order differentiable, by applying the implicit function theorem and taking the first-order derivative of $\boldsymbol{\phi}$,

$$\frac{\partial^2 g(\boldsymbol{\theta}_T(\boldsymbol{\phi}), \boldsymbol{\phi})}{\partial \boldsymbol{\theta}_T^2} \frac{d\boldsymbol{\theta}_T(\boldsymbol{\phi})}{d\boldsymbol{\phi}} + \frac{\partial^2 g(\boldsymbol{\theta}_T(\boldsymbol{\phi}), \boldsymbol{\phi})}{\partial \boldsymbol{\theta}_T \partial \boldsymbol{\phi}} \approx 0. \tag{23}$$

Then, if the Hessian is further assumed to be invertible, the meta gradient can be approximated as

$$\nabla h(\boldsymbol{\phi}) \approx - \underbrace{\frac{\partial f(\boldsymbol{\theta}_T(\boldsymbol{\phi}), \boldsymbol{\phi})}{\partial \boldsymbol{\theta}_T}}_{\boldsymbol{d}} \underbrace{\left(\frac{\partial^2 g(\boldsymbol{\theta}_T(\boldsymbol{\phi}), \boldsymbol{\phi})}{\partial \boldsymbol{\theta}_T^2}\right)^{-1}}_{\boldsymbol{H}^{-1}} \underbrace{\frac{\partial^2 g(\boldsymbol{\theta}_T(\boldsymbol{\phi}), \boldsymbol{\phi})}{\partial \boldsymbol{\theta}_T \partial \boldsymbol{\phi}}}_{\boldsymbol{Y}} + \underbrace{\frac{\partial f(\boldsymbol{\theta}_T(\boldsymbol{\phi}), \boldsymbol{\phi})}{\partial \boldsymbol{\phi}}}_{\boldsymbol{c}}. \tag{24}$$

The main challenge lies in the computation of the inverse Hessian matrix $\boldsymbol{H}^{-1}$, which is intractable when $\boldsymbol{\theta}$ is of high dimensionality. Fortunately, several iterative inverse Hessian vector product (`ihvp`) approximators requiring only Hessian vector product (`hvp`) and $\mathcal{O}(M)$ space can be employed to produce $\widehat{\boldsymbol{dH}^{-1}}$ for approximating $\boldsymbol{dH}^{-1}$, based on Conjugate Gradient [48, 53], Neumann Series [19, 28] and low-rank approximation [64, 24].

**Neumann Series**. The inverse Hessian vector product can be approximated with a truncated sum of Neumann series [19, 28],

$$\widehat{\boldsymbol{dH}^{-1}} = \alpha \sum_{k=0}^{K} \boldsymbol{d}(\boldsymbol{I} - \alpha \boldsymbol{H})^k = \boldsymbol{dH}^{-1} - \alpha \sum_{k=K+1}^{\infty} \boldsymbol{d}(\boldsymbol{I} - \alpha \boldsymbol{H})^k, \tag{25}$$

where $\alpha$ is a hyperparameter to ensure the convergence, and $K$ is the number of truncated steps. Compared to other IF-based methods, the Neumann Series has demonstrated good empirical performance and stability [19], and its stochastic variant has been well studied [28].

**Weakness (IF): Approximation Errors**. The errors of IF emanate from two distinct sources Firstly, IFT presupposes that the Karush-Kuhn-Tucker (KKT) conditions of the inner problem are satisfied, leading to an approximation error in (23) when iterative approximations of the inner solutions are used. Secondly, the singular nature of the Hessian within neural network training [57] leads to costly and unstable inverse Hessian approximation in practical applications, with a heavy reliance on engineering efforts [9]. More formally, recall

$$\nabla h(\boldsymbol{\phi}) = \underbrace{\frac{\partial f(\boldsymbol{\theta}_T(\boldsymbol{\phi}), \boldsymbol{\phi})}{\partial \boldsymbol{\theta}_T}}_{\boldsymbol{d}} \underbrace{\frac{d\boldsymbol{\theta}_T(\boldsymbol{\phi})}{d\boldsymbol{\phi}}}_{\boldsymbol{Z}} + \underbrace{\frac{\partial f(\boldsymbol{\theta}_T(\boldsymbol{\phi}), \boldsymbol{\phi})}{\partial \boldsymbol{\phi}}}_{\boldsymbol{c}}. \tag{26}$$

The approximation error can be decomposed into

$$\underbrace{\nabla h(\boldsymbol{\phi}) - \hat{\nabla} h(\boldsymbol{\phi})}_{\epsilon} = \underbrace{\boldsymbol{d}(\boldsymbol{Z} + \boldsymbol{H}^{-1}\boldsymbol{Y})}_{\epsilon_{\text{if}}} + \underbrace{(\widehat{\boldsymbol{dH}^{-1}} - \boldsymbol{dH}^{-1})\boldsymbol{Y}}_{\epsilon_{\text{inv}}}. \tag{27}$$

To reduce the computational cost, Hessian-free approaches [76, 75, 9] propose approximating the Hessian as an identity matrix, incorporating additional assumptions about the inner model and objective.

$$\widehat{\boldsymbol{dH}^{-1}} = \alpha \boldsymbol{dI}, \tag{28}$$

where $\alpha > 0$ is a hyperparameter to control the magnitude. However, these numerous assumptions often diverge from practical scenarios, resulting in significant approximation errors and consequently inducing suboptimal outcomes.

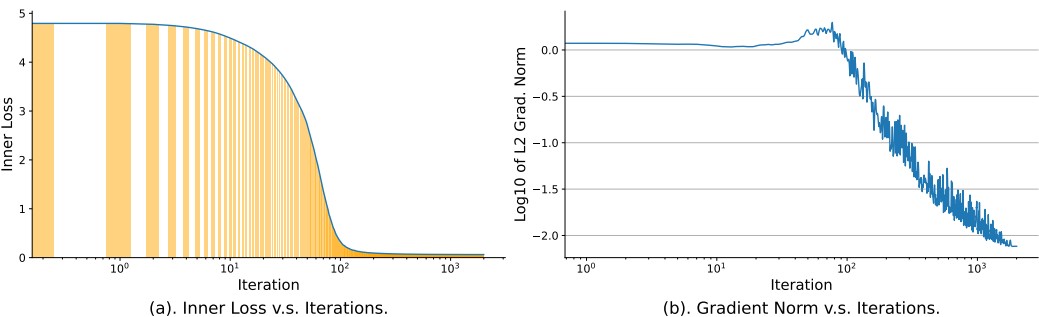

(a). Inner Loss v.s. Iterations.     (b). Gradient Norm v.s. Iterations.

Figure B.1: **CIFAR100, IPC=50**: Inner Loss and gradient norm for Neumann

In Figure B.1, it is evident that the inner optimization has not converged by the unrolled step 100, as indicated by both inner loss and gradient norm. This observation implies that the Karush-Kuhn-Tucker (KKT) conditions are not satisfied, leading to the conclusion that the approximation used in (23) introduces a bias.

## B.3 Value Function (VF)

The VF-based methodology [39, 37, 61, 33] considers an equivalent reformulation of the original optimization problem as outlined in (2):

$$\min_{\boldsymbol{\theta},\boldsymbol{\phi}} \ f(\boldsymbol{\theta},\boldsymbol{\phi}) \quad s.t. \ g(\boldsymbol{\theta},\boldsymbol{\phi}) \leq g(\boldsymbol{\theta}_T(\boldsymbol{\phi}),\boldsymbol{\phi}). \tag{29}$$

This reformulation casts the standard bi-level optimization challenge into a constrained single-level optimization framework. VF-based methods circumvent the need for second-order computations and have demonstrated near-optimal complexity, comparable to second-order methodologies in deterministic settings, as reported in [8].

**Weakness (VF): stochastic optimization**. VF-based strategies have yet to gain widespread acceptance in practical ML applications. This limited adoption is primarily attributed to the challenges these methods face in addressing large-scale stochastic problems, where the complexity significantly impedes their performance [73].

# C Proofs of Theoretical Results

In this section, we detail the proofs of Lemma 3.1 and Theorem 3.4. Our approach to proving Theorem 3.4 follows a similar high-level approach as [19, 28, 60] with some important distinctions. Initially, in Lemma B.2, we extend the smoothness properties of the objective functions $f$ and $g$, and the transition functions $\Omega$, to the $T$-th iteration lower-level parameter $\boldsymbol{\theta}_T$. Following this, Lemma B.3 establishes the smoothness of the meta-learning objective $f(\boldsymbol{\theta}_T, \boldsymbol{\phi})$, incorporating results from the inner-loop computations. Building on the demonstrated smoothness of the meta objective function and the variance of the forward gradient method (as shown in Lemma 3.1), we then validate the convergence properties of Algorithm 1.

The novelty in our proof of Theorem 3.4 lies in two primary aspects. Firstly, our analysis does not presume that the lower-level optimization yields an optimal solution $\boldsymbol{\theta}^*$; instead, it more realistically assumes the use of $\boldsymbol{\theta}_T$, which is derived from a finite number of iterations. This assumption aligns more closely with the computational constraints encountered in real-world scenarios. Secondly, our convergence analysis explicitly accounts for the variance of our unbiased gradient estimator, achieving a convergence rate of $\mathcal{O}(\epsilon^{-1}\rho^{-1})$. This demonstrates that utilizing the forward gradient method, while significantly reducing memory requirements, does not adversely affect the algorithm's convergence rate, underscoring the practical viability and efficiency of our approach even with memory constraints.

In Appendix C.3, we discuss how to extend the convergence of optimization problem (2) into (1), with additional assumptions.

## C.1 Proof of Lemma 3.1

For convenience, the lemma is restated as follows.

**Lemma 3.1.** *For any $\boldsymbol{\phi} \in \Phi$, the gradient estimation with forward gradient method:*

$$\hat{\nabla}h(\boldsymbol{\phi}) = \frac{1}{b}\sum_{i=1}^{b} \nabla h(\boldsymbol{\phi})\boldsymbol{v}_i\boldsymbol{v}_i^{\top},$$

*where $\boldsymbol{v}_i \sim \mathrm{Unif}(\{-1,1\}^N)$, and b denotes the sample size, satifies*

$$\mathbb{E}\|\hat{\nabla}h(\boldsymbol{\phi}) - \nabla h(\boldsymbol{\phi})\|^2 = \frac{1}{\rho}\|\nabla h(\boldsymbol{\phi})\|^2,$$

*where $\rho := \frac{b}{N-1} \in (0,1]$ as b is selected from $1, \ldots, N-1$.*

*Proof.* We start by computing the variance of one-sample estimation, $\hat{\nabla}h(\boldsymbol{\phi}) = \nabla h(\boldsymbol{\phi})\boldsymbol{v}\boldsymbol{v}^{\top}$. Since $\mathbb{E}[\boldsymbol{v}\boldsymbol{v}^{\top}] = \mathbf{I}$, we know that $\mathbb{E}[\hat{\nabla}h(\boldsymbol{\phi})] = \nabla h(\boldsymbol{\phi})$. Consequently,

$$\begin{aligned}
\mathbb{E}\|\hat{\nabla}h(\boldsymbol{\phi}) - \mathbb{E}\hat{\nabla}h(\boldsymbol{\phi})\|^2 &= \mathbb{E}\|\nabla h(\boldsymbol{\phi})(\boldsymbol{v}\boldsymbol{v}^{\top} - \mathbf{I})\|^2 \\
&= \mathbb{E}[\nabla h(\boldsymbol{\phi})^{\top}(\boldsymbol{v}\boldsymbol{v}^{\top} - \mathbf{I})^{\top}(\boldsymbol{v}\boldsymbol{v}^{\top} - \mathbf{I})(\nabla h(\boldsymbol{\phi}))] \\
&= \mathbb{E}[(\nabla h(\boldsymbol{\phi}))^{\top}(\boldsymbol{v}\boldsymbol{v}^{\top})^{\top}\boldsymbol{v}\boldsymbol{v}^{\top}\nabla h(\boldsymbol{\phi}) - 2(\nabla h(\boldsymbol{\phi}))^{\top}(\boldsymbol{v}\boldsymbol{v}^{\top})^{\top}\nabla h(\boldsymbol{\phi}) + (\nabla h(\boldsymbol{\phi}))^{\top}\nabla h(\boldsymbol{\phi})] \\
&= \mathbb{E}\|\nabla h(\boldsymbol{\phi})\boldsymbol{v}\boldsymbol{v}^{\top}\|^2 - 2\mathbb{E}\|\nabla h(\boldsymbol{\phi})\boldsymbol{v}\|^2 + \|\nabla h(\boldsymbol{\phi})\|^2.
\end{aligned} \tag{30}$$

Since $\boldsymbol{v}$ is an $N$-dimensional Rademacher random variable, we have $\mathbb{E}\|\nabla h(\boldsymbol{\phi})\boldsymbol{v}\|^2 = \|\nabla h(\boldsymbol{\phi})\|^2$ and $\mathbb{E}\|\boldsymbol{v}\boldsymbol{v}^{\top}\|^2 = N$. Then,

$$\begin{aligned}
(30) &= \mathbb{E}\|\nabla h(\boldsymbol{\phi})\boldsymbol{v}\boldsymbol{v}^{\top}\|^2 - \|\nabla h(\boldsymbol{\phi})\|^2 \\
&= (N-1)\|\nabla h(\boldsymbol{\phi})\|^2.
\end{aligned}$$

For the multi-sample estimation $\hat{\nabla}h(\phi) = \frac{1}{b}\sum_{i=1}^{b}\nabla h(\phi)v_iv_i^\top$. Since $v_i$ are i.i.d. sampled, and $\mathbb{E}[\hat{\nabla}h(\phi)] = \nabla h(\phi)$, we have

$$\mathbb{E}\|\hat{\nabla}h(\phi) - \mathbb{E}\hat{\nabla}h(\phi)\|^2 = \mathbb{E}\left\|\nabla h(\phi)\frac{1}{b}\sum_{i=1}^{b}(v_iv_i^\top - \mathbf{I})\right\|^2$$

$$= \frac{1}{b^2}\sum_{i=1}^{b}\mathbb{E}\|\nabla h(\phi)(v_iv_i^\top - \mathbf{I})\|^2$$

$$= \frac{N-1}{b}\|\nabla h(\phi)\|^2.$$

$\square$

## C.2 Proof of Theorem 3.4

To prove our main result (Theorem 3.4), we first establish useful smoothness properties of the hyperparameter learned from solving the lower-level optimization problem. Subsequently, we establish the smoothness of the meta objective function examined at the approximated lower-level parameters in Lemma B.3.

Regarding the lower-level parameter $\theta(\phi)$, we present the following lemma, which is based on Assumption 3.3, and establishes that $\theta(\phi)$ inherits similar Lipschitz continuity and smoothness properties as $\Omega_t$.

**Lemma B.2.** *Under Assumptions 3.2 and 3.3, $\theta_T(\phi)$ is $C_Z$-Lipchitz and $L_Z$-smooth, i.e., for any $\phi, \phi' \in \Phi$,*

$$\|\theta_T(\phi) - \theta_T(\phi')\| \leq C_Z\|\phi - \phi'\|, \quad \|\nabla\theta_T(\phi) - \nabla\theta_T(\phi')\| \leq L_Z\|\phi - \phi'\|,$$

*where $C_Z = \frac{C_\Omega^{T+2} - C_\Omega}{C_\Omega - 1}$ and $L_Z = L_\Omega\left[C_\Omega^T + \frac{C_\Omega^{T+2}T}{C_\Omega - 1} - \frac{C_\Omega^T - 1}{(C_\Omega - 1)^2}\right]$.*

*Proof.* We start with the proof of Lipshitz continuity. For any pair of $\phi, \phi' \in \Phi$, using (2) and Assumption 3.3, we have

$$\|\theta_s(\phi) - \theta_s(\phi')\| = \|\Omega(\theta_{s-1}(\phi), \phi) - \Omega(\theta_{s-1}(\phi'), \phi')\|$$
$$\leq C_\Omega\|\theta_{s-1}(\phi) - \theta_{s-1}(\phi')\| + C_\Omega\|\phi - \phi'\|. \tag{31}$$

Applying (31) recursively over $s = 1, \ldots, t$ gives

$$\|\theta_t(\phi) - \theta_t(\phi')\| \leq C_\Omega^t\|\theta_0(\phi) - \theta_0(\phi')\| + \sum_{s=1}^{t}C_\Omega^s\|\phi - \phi'\|$$

$$\leq \sum_{s=1}^{t+1}C_\Omega^s\|\phi - \phi'\| = \frac{C_\Omega^{t+2} - C_\Omega}{C_\Omega - 1}\|\phi - \phi'\|, \tag{32}$$

where the last inequality holds from the fact that $\theta_0 = \Omega_0$ as well as Assumption 3.3, and the subsequent equality follows from the geometric series summation formula.

Therefore, $\theta_t(\phi)$ is $C_Z(t)$-Lipchitz, where $C_Z(t) := \frac{C_\Omega^{t+2} - C_\Omega}{C_\Omega - 1}$. Substituting $t = T$, we get $C_Z = C_Z(T) = \frac{C_\Omega^{T+2} - C_\Omega}{C_\Omega - 1}$.

We now proceed with the proof of $L_Z$-smoothness. For simplicity of notation, we follow (4) and denote

$$Z_t(\phi) = \nabla\theta_t(\phi); \quad A_t(\phi) = \frac{\partial\Omega_t(\theta_{t-1}(\phi), \phi)}{\partial\theta_{t-1}}; \quad B_t(\phi) = \frac{\partial\Omega_t(\theta_{t-1}(\phi), \phi)}{\partial\phi}.$$

Subsequently, considering the update rule $Z_t(\phi) = A_t(\phi)Z_{t-1}(\phi) + B_t(\phi)$ of Forward Gradient Unrolling (4), we have

$$\|\nabla\theta_t(\phi) - \nabla\theta_t(\phi')\|$$
$$= \|Z_t(\phi) - Z_t(\phi')\|$$
$$\overset{(4)}{=} \|A_t(\phi)Z_{t-1}(\phi) + B_t(\phi) - [A_t(\phi')Z_{t-1}(\phi') + B_t(\phi')]\|$$
$$\leq \|A_t(\phi)Z_{t-1}(\phi) - A_t(\phi')Z_{t-1}(\phi')\| + \|B_t(\phi) - B_t(\phi')\|$$
$$= \|A_t(\phi)Z_{t-1}(\phi) - A_t(\phi)Z_{t-1}(\phi') + A_t(\phi)Z_{t-1}(\phi') - A_t(\phi')Z_{t-1}(\phi')\| + \|B_t(\phi) - B_t(\phi')\|$$
$$\leq \|A_t(\phi)Z_{t-1}(\phi) - A_t(\phi)Z_{t-1}(\phi')\| + \|A_t(\phi)Z_{t-1}(\phi') - A_t(\phi')Z_{t-1}(\phi')\|$$
$$\quad + \|B_t(\phi) - B_t(\phi')\|$$
$$\leq \|A_t(\phi)\| \cdot \|Z_{t-1}(\phi) - Z_{t-1}(\phi')\| + \|A_t(\phi) - A_t(\phi')\| \cdot \|Z_{t-1}(\phi')\| + \|B_t(\phi) - B_t(\phi')\|$$
$$\leq C_\Omega\|Z_{t-1}(\phi) - Z_{t-1}(\phi')\| + (C_Z(t) + 1)L_\Omega\|\phi - \phi'\|, \tag{33}$$

where the last inequality follows from Assumption 3.3 that $\boldsymbol{\Omega}_0(\boldsymbol{\phi})$ and $\boldsymbol{\Omega}_{1:T}(\boldsymbol{\psi})$ are $C_{\boldsymbol{\Omega}}$-Lipshitz and $L_{\boldsymbol{\Omega}}$-smooth, and the previously proved result that $\boldsymbol{\theta}_t(\boldsymbol{\phi})$ is $C_{\boldsymbol{Z}}(t)$-Lipchitz.

Noting that $\boldsymbol{Z}_0(\boldsymbol{\phi}) = \nabla\boldsymbol{\theta}_0(\boldsymbol{\phi}) = \nabla\boldsymbol{\Omega}_0(\boldsymbol{\phi})$, applying (33) recursively over $t = 1, \ldots, T$ gives

$$\|\nabla\boldsymbol{\theta}_T(\boldsymbol{\phi}) - \nabla\boldsymbol{\theta}_T(\boldsymbol{\phi}')\| \leq C_{\boldsymbol{\Omega}}^T\|\boldsymbol{Z}_0(\boldsymbol{\phi}) - \boldsymbol{Z}_0(\boldsymbol{\phi}\text{'})\| + \sum_{t=1}^{T} C_{\boldsymbol{\Omega}}^{T-t}(C_{\boldsymbol{Z}}(t)+1)L_{\boldsymbol{\Omega}}\|\boldsymbol{\phi}-\boldsymbol{\phi}'\|$$

$$= C_{\boldsymbol{\Omega}}^T\|\nabla\boldsymbol{\Omega}_0(\boldsymbol{\phi}) - \nabla\boldsymbol{\Omega}_0(\boldsymbol{\phi}')\| + C_{\boldsymbol{\Omega}}^T\sum_{t=1}^{T}\frac{C_{\boldsymbol{Z}}(t)+1}{C_{\boldsymbol{\Omega}}^t}L_{\boldsymbol{\Omega}}\|\boldsymbol{\phi}-\boldsymbol{\phi}'\|$$

$$\leq L_{\boldsymbol{\Omega}}C_{\boldsymbol{\Omega}}^T\|\boldsymbol{\phi}-\boldsymbol{\phi}'\| + C_{\boldsymbol{\Omega}}^T\sum_{t=1}^{T}\frac{C_{\boldsymbol{\Omega}}^{t+2}-1}{C_{\boldsymbol{\Omega}}^t(C_{\boldsymbol{\Omega}}-1)}L_{\boldsymbol{\Omega}}\|\boldsymbol{\phi}-\boldsymbol{\phi}'\|$$

$$= L_{\boldsymbol{\Omega}}\left[C_{\boldsymbol{\Omega}}^T + C_{\boldsymbol{\Omega}}^T\sum_{t=1}^{T}\frac{C_{\boldsymbol{\Omega}}^2}{C_{\boldsymbol{\Omega}}-1} - \frac{C_{\boldsymbol{\Omega}}^T}{C_{\boldsymbol{\Omega}}-1}\sum_{t=1}^{T}\frac{1}{C_{\boldsymbol{\Omega}}^t}\right]\|\boldsymbol{\phi}-\boldsymbol{\phi}'\|$$

$$= L_{\boldsymbol{\Omega}}\left[C_{\boldsymbol{\Omega}}^T + \frac{C_{\boldsymbol{\Omega}}^{T+2}T}{C_{\boldsymbol{\Omega}}-1} - \frac{C_{\boldsymbol{\Omega}}^T}{C_{\boldsymbol{\Omega}}-1}\frac{\frac{1}{C_{\boldsymbol{\Omega}}}(1-\frac{1}{C_{\boldsymbol{\Omega}}^T})}{1-\frac{1}{C_{\boldsymbol{\Omega}}}}\right]\|\boldsymbol{\phi}-\boldsymbol{\phi}'\|$$

$$= L_{\boldsymbol{\Omega}}\left[C_{\boldsymbol{\Omega}}^T + \frac{C_{\boldsymbol{\Omega}}^{T+2}T}{C_{\boldsymbol{\Omega}}-1} - \frac{C_{\boldsymbol{\Omega}}^T-1}{(C_{\boldsymbol{\Omega}}-1)^2}\right]\|\boldsymbol{\phi}-\boldsymbol{\phi}'\|,$$

where the third line follows from Assumption 3.3 that $\boldsymbol{\Omega}_0(\boldsymbol{\phi})$ is $L_{\boldsymbol{\Omega}}$-smooth and the choice $C_{\boldsymbol{Z}}(t) = \frac{C_{\boldsymbol{\Omega}}^{t+2}-C_{\boldsymbol{\Omega}}}{C_{\boldsymbol{\Omega}}-1}$ (which gives $C_{\boldsymbol{Z}}(t) + 1 = \frac{C_{\boldsymbol{\Omega}}^{t+2}-1}{C_{\boldsymbol{\Omega}}-1}$), and the fifth line again uses the geometric series summation formula.

Hence, $\boldsymbol{\theta}_T(\boldsymbol{\phi})$ is $L_{\boldsymbol{Z}}$-smooth with $L_{\boldsymbol{Z}} = L_{\boldsymbol{\Omega}}\left[C_{\boldsymbol{\Omega}}^T + \frac{C_{\boldsymbol{\Omega}}^{T+2}T}{C_{\boldsymbol{\Omega}}-1} - \frac{C_{\boldsymbol{\Omega}}^T-1}{(C_{\boldsymbol{\Omega}}-1)^2}\right].$  □

Next, we provide a lemma establishing that the upper-level objective $f$, evaluated at the learned parameter $(\boldsymbol{\theta}_T(\boldsymbol{\phi}), \boldsymbol{\phi})$, also adheres to certain smoothness properties.

**Lemma B.3.** *Define $h(\boldsymbol{\phi}) := f(\boldsymbol{\theta}_T(\boldsymbol{\phi}), \boldsymbol{\phi})$. Under Assumptions 3.2 and 3.3, $h(\boldsymbol{\phi})$ is $L_h$-smooth, i.e., for any $\boldsymbol{\phi}, \boldsymbol{\phi}' \in \Phi$,*

$$\|\nabla h(\boldsymbol{\phi}) - \nabla h(\boldsymbol{\phi}')\| \leq L_h\|\boldsymbol{\phi} - \boldsymbol{\phi}'\|,$$

*where $L_h = (C_{\boldsymbol{Z}} + 1)^2 L + C L_{\boldsymbol{Z}}$, with $C_{\boldsymbol{Z}}$ and $L_{\boldsymbol{Z}}$ defined in Lemma B.2.*

*Proof.* For simplicity of notation, we follow (5) and denote

$$\boldsymbol{Z}_t(\boldsymbol{\phi}) = \nabla\boldsymbol{\theta}_t(\boldsymbol{\phi}); \quad \boldsymbol{c}_T(\boldsymbol{\phi}) = \frac{\partial f(\boldsymbol{\theta}_T(\boldsymbol{\phi}), \boldsymbol{\phi})}{\partial\boldsymbol{\phi}}; \quad \boldsymbol{d}_T(\boldsymbol{\phi}) = \frac{\partial f(\boldsymbol{\theta}_T(\boldsymbol{\phi}), \boldsymbol{\phi})}{\partial\boldsymbol{\theta}_T}.$$

For any $\boldsymbol{\phi}, \boldsymbol{\phi}' \in \Phi$, following a similar proof as Lemma B.2, we have

$$\|\nabla h(\boldsymbol{\phi}) - \nabla h(\boldsymbol{\phi}')\|$$
$$\overset{(5)}{=} \|\boldsymbol{d}_T(\boldsymbol{\phi})\boldsymbol{Z}_T(\boldsymbol{\phi}) + \boldsymbol{c}(\boldsymbol{\phi}) - (\boldsymbol{d}_T(\boldsymbol{\phi}')\boldsymbol{Z}_T(\boldsymbol{\phi}') + \boldsymbol{c}(\boldsymbol{\phi}'))\|$$
$$\leq \|\boldsymbol{d}_T(\boldsymbol{\phi})\boldsymbol{Z}_T(\boldsymbol{\phi}) - \boldsymbol{d}_T(\boldsymbol{\phi}')\boldsymbol{Z}_T(\boldsymbol{\phi}')\| + \|\boldsymbol{c}_T(\boldsymbol{\phi}) - \boldsymbol{c}_T(\boldsymbol{\phi}')\| \qquad (34)$$
$$= \|\boldsymbol{d}_T(\boldsymbol{\phi})\boldsymbol{Z}_T(\boldsymbol{\phi}) - \boldsymbol{d}_T(\boldsymbol{\phi}')\boldsymbol{Z}_T(\boldsymbol{\phi}) + \boldsymbol{d}_T(\boldsymbol{\phi}')\boldsymbol{Z}_T(\boldsymbol{\phi}) - \boldsymbol{d}_T(\boldsymbol{\phi}')\boldsymbol{Z}_T(\boldsymbol{\phi}')\| + \|\boldsymbol{c}_T(\boldsymbol{\phi}) - \boldsymbol{c}_T(\boldsymbol{\phi}')\|$$
$$\leq \|\boldsymbol{d}_T(\boldsymbol{\phi}) - \boldsymbol{d}_T(\boldsymbol{\phi}')\| \cdot \|\boldsymbol{Z}_T(\boldsymbol{\phi})\| + \|\boldsymbol{d}_T(\boldsymbol{\phi}')\| \cdot \|\boldsymbol{Z}_T(\boldsymbol{\phi}) - \boldsymbol{Z}_T(\boldsymbol{\phi}')\| + \|\boldsymbol{c}_T(\boldsymbol{\phi}) - \boldsymbol{c}_T(\boldsymbol{\phi}')\|$$

Subsequently, we deduce that

$$(34) \leq C_{\boldsymbol{Z}}\|\boldsymbol{d}_T(\boldsymbol{\phi}) - \boldsymbol{d}_T(\boldsymbol{\phi}')\| + CL_{\boldsymbol{Z}}\|\boldsymbol{\phi}-\boldsymbol{\phi}'\| + \|\boldsymbol{c}_T(\boldsymbol{\phi}) - \boldsymbol{c}_T(\boldsymbol{\phi}')\|$$
$$\leq C_{\boldsymbol{Z}}\left(\left\|\frac{\partial f(\boldsymbol{\theta}_T(\boldsymbol{\phi}), \boldsymbol{\phi})}{\partial\boldsymbol{\theta}_T} - \frac{\partial f(\boldsymbol{\theta}_T(\boldsymbol{\phi}'), \boldsymbol{\phi})}{\partial\boldsymbol{\theta}_T}\right\| + \left\|\frac{\partial f(\boldsymbol{\theta}_T(\boldsymbol{\phi}'), \boldsymbol{\phi})}{\partial\boldsymbol{\theta}_T} - \frac{\partial f(\boldsymbol{\theta}_T(\boldsymbol{\phi}'), \boldsymbol{\phi}')}{\partial\boldsymbol{\theta}_T}\right\|\right)$$
$$+ \left(\left\|\frac{\partial f(\boldsymbol{\theta}_T(\boldsymbol{\phi}), \boldsymbol{\phi})}{\partial\boldsymbol{\phi}} - \frac{\partial f(\boldsymbol{\theta}_T(\boldsymbol{\phi}'), \boldsymbol{\phi})}{\partial\boldsymbol{\phi}}\right\| + \left\|\frac{\partial f(\boldsymbol{\theta}_T(\boldsymbol{\phi}'), \boldsymbol{\phi})}{\partial\boldsymbol{\phi}} - \frac{\partial f(\boldsymbol{\theta}_T(\boldsymbol{\phi}'), \boldsymbol{\phi}')}{\partial\boldsymbol{\phi}'}\right\|\right)$$
$$+ CL_{\boldsymbol{Z}}\|\boldsymbol{\phi}-\boldsymbol{\phi}'\|$$
$$\leq C_{\boldsymbol{Z}}L\|\boldsymbol{\theta}_T(\boldsymbol{\phi}) - \boldsymbol{\theta}_T(\boldsymbol{\phi}')\| + C_{\boldsymbol{Z}}L\|\boldsymbol{\phi}-\boldsymbol{\phi}'\| + L\|\boldsymbol{\theta}_T(\boldsymbol{\phi}) - \boldsymbol{\theta}_T(\boldsymbol{\phi}')\| + L\|\boldsymbol{\phi}-\boldsymbol{\phi}'\|$$
$$+ CL_{\boldsymbol{Z}}\|\boldsymbol{\phi}-\boldsymbol{\phi}'\|$$
$$\leq C_{\boldsymbol{Z}}LC_{\boldsymbol{Z}}\|\boldsymbol{\phi}-\boldsymbol{\phi}'\| + C_{\boldsymbol{Z}}L\|\boldsymbol{\phi}-\boldsymbol{\phi}'\| + LC_{\boldsymbol{Z}}\|\boldsymbol{\phi}-\boldsymbol{\phi}'\| + L\|\boldsymbol{\phi}-\boldsymbol{\phi}'\| + CL_{\boldsymbol{Z}}\|\boldsymbol{\phi}-\boldsymbol{\phi}'\|$$
$$= [(C_{\boldsymbol{Z}}+1)^2 L + CL_{\boldsymbol{Z}}]\|\boldsymbol{\phi}-\boldsymbol{\phi}'\|,$$

where the first, third, and fourth lines all follow directly from Lemma B.2 and Assumption 3.3 (recall that the latter states that $f$ is $L$-Lipschitz and $C$-smooth).

Therefore, $h(\boldsymbol{\phi})$ is $L_h$-smooth with $L_h = (C_{\boldsymbol{Z}} + 1)^2 L + C L_{\boldsymbol{Z}}$. $\qquad\square$

Now based on the aforementioned lemmas, we put forward the proof of our main theorem: the convergence analysis for our bilevel optimization method $\textbf{(FG)}^{\textbf{2}}\textbf{U}$.

**Theorem 3.4** (Convergence). *Suppose that Asumption 3.2 and Assumption 3.3 hold. Setting the learning rate $\beta = \frac{\rho}{(\rho+1)L_h}$ for gradient descent over the hyperparameter $\boldsymbol{\phi}$, we have*

$$\frac{1}{K}\sum_{k=0}^{K-1}\mathbb{E}\left[\|\nabla h(\boldsymbol{\phi}_k)\|^2\right] \leq \frac{4L_h\left(\mathbb{E}[h(\boldsymbol{\phi}_0)] - \min_{\boldsymbol{\phi}} h(\boldsymbol{\phi})\right)}{\rho K}. \tag{35}$$

*Proof.* We have

$$
\begin{aligned}
& h(\boldsymbol{\phi}_{k+1}) - h(\boldsymbol{\phi}_k) \\
& \leq \langle \nabla h(\boldsymbol{\phi}_k), \boldsymbol{\phi}_{k+1} - \boldsymbol{\phi}_k \rangle + \frac{L_h}{2}\|\boldsymbol{\phi}_{k+1} - \boldsymbol{\phi}_k\|^2 \\
& = -\beta\langle \nabla h(\boldsymbol{\phi}_k), \hat{\nabla} h(\boldsymbol{\phi}_k) \rangle + \frac{\beta^2 L_h}{2}\|\hat{\nabla} h(\boldsymbol{\phi}_k)\|^2 \\
& = -\beta\langle \nabla h(\boldsymbol{\phi}_k), \hat{\nabla} h(\boldsymbol{\phi}_k) \rangle + \frac{\beta^2 L_h}{2}\|\nabla h(\boldsymbol{\phi}_k) + \hat{\nabla} h(\boldsymbol{\phi}_k) - \nabla h(\boldsymbol{\phi}_k)\|^2 \\
& = -\frac{\beta^2 L_h}{2}\|\nabla h(\boldsymbol{\phi}_k)\|^2 + (\beta^2 L_h - \beta)\langle \nabla h(\boldsymbol{\phi}_k), \hat{\nabla} h(\boldsymbol{\phi}_k) \rangle + \frac{\beta^2 L_h}{2}\|\nabla h(\boldsymbol{\phi}_k) - \hat{\nabla} h(\boldsymbol{\phi}_k)\|^2,
\end{aligned}
\tag{36}
$$

where the second line is a well-known inequality for smooth functions with the $L_h$-smoothness itself following from Lemma B.3, and the third line uses the gradient descent rule $\boldsymbol{\phi}_{k+1} = \boldsymbol{\phi}_k - \beta\hat{\nabla} h(\boldsymbol{\phi}_k)$.

By Lemma 3.1 and the fact that $\hat{\nabla} h$ is unbiased (see the proof of Lemma 3.1), we know that

$$\mathbb{E}[\langle \nabla h(\boldsymbol{\phi}_k), \hat{\nabla} h(\boldsymbol{\phi}_k)\rangle | \boldsymbol{\phi}_k] = \|\nabla h(\boldsymbol{\phi}_k)\|^2;$$
$$\mathbb{E}[\|\nabla h(\boldsymbol{\phi}_k) - \hat{\nabla} h(\boldsymbol{\phi}_k)\|^2 | \boldsymbol{\phi}_k] = \frac{1}{\rho}\|\nabla h(\boldsymbol{\phi}_k)\|^2.$$

Therefore, taking the conditional expectation $\mathbb{E}[\,\cdot\,|\,\boldsymbol{\phi}_k]$ over (36) gives

$$\mathbb{E}[h(\boldsymbol{\phi}_{k+1})|\boldsymbol{\phi}_k] - h(\boldsymbol{\phi}_k) \leq -\left[\beta - \left(1 + \frac{1}{\rho}\right)\frac{\beta^2 L_h}{2}\right]\|\nabla h(\boldsymbol{\phi}_k)\|^2. \tag{37}$$

Furthermore, taking the full expectation and telescoping (37) over $k$ form 0 to $K-1$ yields

$$
\begin{aligned}
\frac{1}{K}\sum_{k=0}^{K-1}\left[\beta - \frac{(\rho+1)L_h}{2\rho}\beta^2\right]\mathbb{E}\left[\|\nabla h(\boldsymbol{\phi}_k)\|^2\right] & \leq \frac{\mathbb{E}[h(\boldsymbol{\phi}_0)] - \mathbb{E}[h(\boldsymbol{\phi}_K)]}{K} \\
& \leq \frac{\mathbb{E}[h(\boldsymbol{\phi}_0)] - \min_{\boldsymbol{\phi}} h(\boldsymbol{\phi})}{K}.
\end{aligned}
\tag{38}
$$

Choosing $\beta = \frac{\rho}{(\rho+1)L_h}$, we have

$$
\begin{aligned}
\frac{1}{K}\sum_{k=0}^{K-1}\mathbb{E}\left[\|\nabla h(\boldsymbol{\phi}_k)\|^2\right] & \leq \frac{2(\rho+1)L_h\left(\mathbb{E}[h(\boldsymbol{\phi}_0)] - \min_{\boldsymbol{\phi}} h(\boldsymbol{\phi})\right)}{\rho K} \\
& \leq \frac{4L_h\left(\mathbb{E}[h(\boldsymbol{\phi}_0)] - \min_{\boldsymbol{\phi}} h(\boldsymbol{\phi})\right)}{\rho K}.
\end{aligned}
\tag{39}
$$

Hence, Algorithm 1 requires $\mathcal{O}(\epsilon^{-1}\rho^{-1})$ steps to attain an $\epsilon$-accurate stationary point. $\qquad\square$

## C.3 Extended Discussions

**Convergence of Problem** (1). To extend the convergence of optimization problem (2) into (1), we need to assume that the lower-level objective function $g$ is strongly convex w.r.t. $\boldsymbol{\theta}$ as commonly done by previous works [60, 28]. From the strong convexity and first-order smoothness (Assumption 3.2) of $g$, we have 1) the zeroth and first-order smoothness of $\boldsymbol{\theta}^*(\boldsymbol{\phi})$; 2) $\|\boldsymbol{\theta}_T(\boldsymbol{\phi}') - \boldsymbol{\theta}^*(\boldsymbol{\phi}')\| \to 0$ as $T \to +\infty$. Then the inequality

$$\|\boldsymbol{\theta}_T(\boldsymbol{\phi}) - \boldsymbol{\theta}_T(\boldsymbol{\phi}')\| \leq \|\boldsymbol{\theta}_T(\boldsymbol{\phi}) - \boldsymbol{\theta}^*(\boldsymbol{\phi})\| + \|\boldsymbol{\theta}_T(\boldsymbol{\phi}') - \boldsymbol{\theta}^*(\boldsymbol{\phi}')\| + \|\boldsymbol{\theta}^*(\boldsymbol{\phi}) - \boldsymbol{\theta}^*(\boldsymbol{\phi}')\| \tag{40}$$

implies the the zeroth and first-order smoothness of $\boldsymbol{\theta}_T(\boldsymbol{\phi})$. Following the same line of proof as presented in our paper, we derive the smoothness of $f(\boldsymbol{\theta}^*(\boldsymbol{\phi}), \boldsymbol{\phi})$ and $f(\boldsymbol{\theta}_T(\boldsymbol{\phi}), \boldsymbol{\phi})$, and subsequently the convergence of either problem (1) or (2).

However, as discussed in Section 2, given that the scope of this paper is large-scale BO, the inner optimization typically involves deep neural networks. Therefore, the optimal parameters are not explicitly accessible and can only be estimated through iterative procedures. Most related works [9, 60, 28] are implicitly or explicitly solving (2) instead of (1). Additionally, it is important to acknowledge that achieving strong convexity is often unfeasible in practical applications. Consequently, we focus on (2), aiming to present a more practical convergence theory that proves the effectiveness of our method.

# D  Zeroth-Order Derivative Estimator

In this section, we give a more detailed introduction to zeroth-order (ZO) derivative estimators. These estimators are pivotal in scenarios where the computation of exact derivatives is either infeasible due to memory constraints or computationally prohibitive. Apart from the forward gradient method employed in $\textbf{(FG)}^2\textbf{U}$, randomized smoothing (RS) is another widely-used derivative estimator, both in Reinforcement Learning [22, 32] and Large Language Models [18, 47, 74].

For a function $F : \mathbb{R}^n \to \mathbb{R}$, gradient estimation via RS can be mathematically formulated as:

$$\nabla_{\boldsymbol{x}} F \approx \mathbb{E}_{v \sim \mathcal{N}(0, I)} \left[ \frac{F(\boldsymbol{x} + \epsilon v) - F(\boldsymbol{x})}{\epsilon} v^\top \right] \approx \frac{1}{b} \sum_{i=1}^{b} \frac{F(\boldsymbol{x} + \epsilon v_i) - F(\boldsymbol{x})}{\epsilon} v_i^\top, \tag{41}$$

where $b$ is the number of random samples, $\epsilon$ is the smoothing parameter, $v_i$ are samples drawn from a standard Gaussian distribution. Regarding the accuracy of estimation, it has been shown in [13, 42] that the variance of RS is roughly in the order of $O(N/b)$, which is the same as FG as proved in Lemma 3.1.

RS stands out particularly in its ability to estimate gradients of functions evaluated through black-box systems, where internal operations are inaccessible or highly complex. This characteristic makes RS exceptionally valuable in practical applications such as adversarial robustness and black-box optimization, where obtaining direct gradients might not be possible. Another advantage of RS is its robustness against noise and discontinuities in the function landscape. Unlike deterministic methods, the stochastic nature of RS allows it to approximate the gradient over a smoothed version of the function, providing stability in scenarios where slight perturbations can lead to substantial changes in the output.

While RS provides robust gradient estimates across various scenarios, it is critical to recognize that RS inherently introduces bias if the expectation is not computed during inference. In many CV and NLP applications, the computational expense of Monte Carlo sampling at the evaluation stage is prohibitive, leading to a biased estimation when using RS. However, in the context of inverse PDE problems, where the inner-loop solvers are non-differentiable numerical solvers, we employ RS as a zeroth-order derivative estimator.

# E  Detailed Task Description

## E.1  Data Condensation

In the era of rapid advancement in machine learning, a multitude of foundation models [11, 6, 55] has benefited from training on large-scale datasets, exhibiting formidable performance that models trained on small-scale data cannot match. However, the exponential growth of data also presents challenges: (1) Models updated with only new data are prone to catastrophic forgetting [20] while retaining all historical data for subsequent training imposes significant storage and computational burdens. (2) Applications within the realm of meta-learning, such as hyperparameter tuning [46, 43] and neural architecture search [78, 38], necessitate multiple training iterations over datasets. The computational cost of these operations scales dramatically with the size of the datasets, posing a bottleneck for efficiency and scalability. (3) The widespread dissemination and utilization of datasets have raised significant concerns regarding privacy and copyright [12].

To overcome the challenges posed by large-scale datasets, a line of work known as data condensation [68, 72] has been proposed, with the idea to generate a compact, synthesized dataset, designed to elicit similar behaviors in machine learning models as those trained with the original, massive dataset. The objectives of the mainstream principles [72] designed for data condensation can be naturally formulated as a bi-level optimization problem. We focus on the best-known principle *performance matching* [72] on classification task, which can be formulated as,

$$\min_{\mathcal{D}_o} \mathcal{L}(\theta_T; \mathcal{D}_o), \quad \text{where } \theta_t = \theta_{t-1} - \eta \nabla \mathcal{L}(\theta_{t-1}; \mathcal{D}_c), \ t = 1, \ldots, T, \tag{42}$$

We conduct our experiments to condense the following image datasets:

- **MNIST** [35]: a handwritten digits dataset containing $60,000$ training images and $10,000$ testing images with the size of $28 \times 28$ from 10 categories.
- **CIFAR 10/100** [31]: colored natural images datasets contraining $50,000$ training images and $10,000$ testing images from $10/100$ categories, respectively.

The scale of the condensed dataset will fundamentally impact the results. Therefore, we consider different scales for each dataset, with images per class set to 1, 10, and 50. The condensed dataset will be used to train random initialized models, and evaluated on a test dataset.

## E.2 Meta Learning Online Adaptation of Language Models

The online adaptation of language models (LM) [34, 27] has been studied recently to keep the knowledge of LM updated to date. However, trivial auto-regressive fine-tuning the LM with uniform weights for all tokens results in poor performance in downstream tasks, as the default average negative log-likelihood (NLL) loss does not accurately reflect the importance of tokens [25]. To address the issue, [25] proposed Context-aware Meta-learned Loss Scaling (CaMeLS) to meta-learning the weights of tokens for effective online adaption. More formally, let $\boldsymbol{\theta}$ denote the parameter of the base model for adaptation, $\boldsymbol{\phi}$ denote the parameter of a parametric weight model to assign weights for each token, the meta-learning online adaption of LM can be formulated as the following bi-level optimization,

$$\min_{\boldsymbol{\phi}} \ \mathcal{L}_{meta}(\boldsymbol{\theta}_T(\boldsymbol{\phi}), \boldsymbol{\phi}) \quad s.t. \ \boldsymbol{\theta}_t(\boldsymbol{\phi}) = \boldsymbol{\theta}_{t-1}(\boldsymbol{\phi}) - \eta \nabla_{\boldsymbol{\theta}} \mathcal{L}_{train}(\boldsymbol{\theta}_{t-1}, w_{\boldsymbol{\phi}}), \ t = 1, \ldots, T. \tag{43}$$

We follow the setting studied by [25], where the downstream task is question-answering. The meta-objective consists of a question-answering term measuring the performance gained from adaptation, and a locality term that prevents the updated base model parameters from excessively changing the base model's behavior. Let $\mathcal{D}_{QA}$ denotes the question-answering dataset, $\mathcal{D}_{loc}$ denotes the locality dataset, and $c \in \mathbb{R}^+$ denotes the weight of the locality term, then the meta objective is formally defined as

$$\mathcal{L}_{meta}(\boldsymbol{\theta}_T(\boldsymbol{\phi}), \boldsymbol{\phi}) := \mathbb{E}_{q,a \sim \mathcal{D}_{QA}} - \log p_{\boldsymbol{\theta}_T}(a|q) + c \mathbb{E}_{x \sim \mathcal{D}_{loc}} \sum_i \mathrm{KL}(p_{\boldsymbol{\theta}_T}(\cdot|x_{:i}) \parallel p_{\boldsymbol{\theta}_0}(\cdot|x_{:i})). \tag{44}$$

The inner objective is defined as a weighted NLL loss, where the weights are determined by the weight model $w_{\boldsymbol{\phi}}$,

$$\mathcal{L}_{train}(\boldsymbol{\theta}, w_{\boldsymbol{\phi}}) := \mathbb{E}_{x \sim \mathcal{D}_{train}} \sum_i -w_{\boldsymbol{\phi}}(x_i, x) \log p_{\boldsymbol{\theta}}(x_i|x_{:i}). \tag{45}$$

The trained weight model is then fine-tuned on unseen online documents and evaluated on corresponding question-answering tasks.

## E.3 Data-driven Discovery of Partial Differential Equations (PDEs)

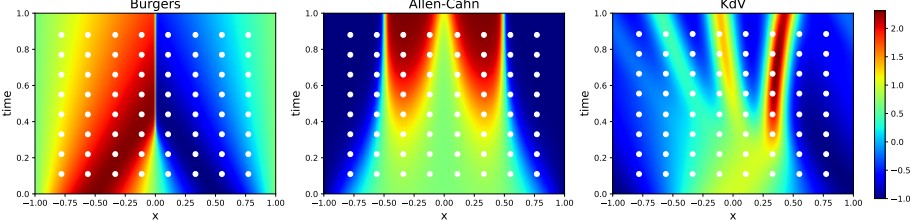

Figure E.1: Visualization of the 2D latent solutions for the Burgers, Allen-Cahn, and KdV equations. The observed data are sampled on an $8 \times 8$ grid, denoted by white points.

We conducted experiments on three non-linear PDEs, with the latent solutions visualized in Figure E.1. The PDE structures (46) (47) (48) are assumed to be known while the PDE parameters $\nu$ in (46) (47) (48) are assumed to be unknown. For each equation, 64 observed data points are sampled on an $8 \times 8$ grid. The objective is to predict the unknown PDE parameters using the observed data. The predicted PDE parameters are evaluated by comparing the error with the ground truth, as well as the error in the corresponding prediction of the latent solution.

### E.3.1 Burgers Equation

The nonlinear viscous Burgers equation is a pivotal partial differential equation arising in diverse domains of applied mathematics such as fluid mechanics, nonlinear acoustics, and traffic flow. This equation can be deduced

from the Navier-Stokes equations for the velocity field by omitting the pressure gradient term. In our experiment, the equation along with Dirichlet boundary conditions, is expressed as follows:

$$
\begin{aligned}
& u_t + u u_x - \nu u_{xx} = 0, \ x \in [-1, 1], t \in [0, 1], \nu > 0, \\
& u(0, x) = -\sin(\pi x), \\
& u(t, -1) = u(t, 1) = 0,
\end{aligned}
\tag{46}
$$

with actual viscosity $\nu = \frac{0.01}{\pi} \approx 0.0031831$. For PINN, following [44], we enforced the initial condition into the output by choosing a surrogate model of the solution as

$$
\hat{u}(x) = (1 - \exp(-t)) \, \mathrm{NN}(x; \boldsymbol{\theta}) - \sin(\pi x),
$$

where $\mathrm{NN}(x; \boldsymbol{\theta})$ is a neural network.

### E.3.2 Allen-Cahn Equation

The Allen-Cahn equation is a reaction-diffusion equation of mathematical physics describing the process of phase separation in multi-component alloy systems, including order-disorder transitions. In our experiment, it is expressed as follows:

$$
\begin{aligned}
& u_t - \nu u_{xx} = 5(u - u^3), x \in [-1, 1], t \in [0, 1], \nu > 0, \\
& u(0, x) = x^2 \cos(\pi x), \\
& u(t, -1) = u(t, 1) = -1,
\end{aligned}
\tag{47}
$$

with the actual diffusion coefficient $\nu = 0.001$. For PINN, we enforced the initial condition into the output by choosing a surrogate model of the solution as

$$
\hat{u}(x) = (1 - \exp(-t)) \, \mathrm{NN}(x; \boldsymbol{\theta}) + x^2 \cos(\pi x),
$$

where $\mathrm{NN}(x; \boldsymbol{\theta})$ is a neural network.

### E.3.3 Korteweg–De Vries (KdV) Equation

The Korteweg–de Vries (KdV) equation serves as a mathematical model for waves on shallow water surfaces. This equation is distinguished as a prototypical example of an integrable PDE. It is characterized by features typical of integrable systems, including a plethora of explicit solutions, notably soliton solutions, and an infinite number of conserved quantities. These properties are particularly noteworthy given the inherent nonlinearity of the equation, which generally complicates the solvability of PDEs. In specific, we consider:

$$
\begin{aligned}
& u_t + u u_x + \nu u_{xxx} = 0, \\
& x \in [-1, 1], t \in [0, 1], \nu \neq 0, \\
& u(0, x) = \cos(\pi x),
\end{aligned}
\tag{48}
$$

with the actual coefficient of dispersion $\nu$ equal to 0.0025. For PINN, we enforced the initial condition into the output by choosing a surrogate model of the solution as

$$
\hat{u}(x) = (1 - \exp(-t)) \, \mathrm{NN}(x; \boldsymbol{\theta}) + \cos(\pi x),
$$

where $\mathrm{NN}(x; \boldsymbol{\theta})$ is a neural network.

### E.3.4 Numerical PDE solver

Numerical solvers play a critical role in the study and application of PDEs, enabling the simulation and analysis of complex physical phenomena that cannot be addressed analytically [51]. These solvers convert PDEs into a form that can be handled computationally, typically by discretizing the domain into a finite set of points or elements and approximating the derivatives. Conventional numerical methods include finite difference methods, finite element methods, and spectral methods [1].

Among the various numerical methods for solving PDEs, spectral methods stand out for their ability to deliver highly accurate solutions, particularly for problems with smooth solutions [21, 7]. Spectral methods involve representing the solution to a PDE as a sum of basis functions, such as trigonometric polynomials, which are globally defined over the domain. This approach contrasts with finite difference or finite element methods, where the solution is localized to the grid points or elements. In this paper, we mainly adopt spectral methods, as we focus on the Burgers, Allen-Cahn, and KdV equations. All these three equations can be efficiently and accurately resolved by spectral techniques.

# F  Implementation Details

## F.1  Data Condensation

We conducted our experiments following the standard data condensation setting established by [68, 77, 67]. The condensation and evaluation are both performed on a depth-3 convolutional neural network [58]. The hyperparameters we used for $(FG)^2U$ are summarized in Appendix F.1. All experiments are conducted on NVIDIA-L40S (40G).

| Datasets | MNIST | | | CIFAR10 | | | CIFAR100 | | |
|---|---|---|---|---|---|---|---|---|---|
| IPC | 1 | 10 | 50 | 1 | 10 | 50 | 1 | 10 | 50 |
| Unrolled Depth | 100 | 100 | 100 | 100 | 100 | 100 | 100 | 100 | 200 |
| # Random Directions | 32 | 32 | 32 | 32 | 32 | 32 | 32 | 32 | 32 |
| # Inner Batch Size | Full | Full | Full | Full | Full | Full | Full | Full | 100 |
| Hessian-Free Pretraining | ✗ | ✓ | ✓ | ✗ | ✓ | ✓ | ✗ | ✓ | ✓ |
| Gradient Accumulate | 16 | 32 | 32 | 32 | 32 | 32 | 64 | 64 | 64 |
| Outer Steps | 10000 | 10000 | 10000 | 10000 | 10000 | 10000 | 10000 | 10000 | 10000 |
| Outer Step Size | 1e-2 | 5e-4 | 5e-4 | 1e-2 | 5e-4 | 5e-4 | 1e-2 | 5e-4 | 5e-4 |
| Evaluation Steps | 1000 | 10000 | 10000 | 1000 | 10000 | 10000 | 1000 | 10000 | 10000 |

Table F.1: $(FG)^2U$ hyperparameters for data condensation experiments.

## F.2  Meta Learning Online Adaptation of Language Models

We adhered to the standard settings of CaMeLS [25] and adapted their official code for our implementation. The only modification made was replacing the meta gradient approximation module with $(FG)^2U$. It is important to note that the base models used for meta-learning were initially pre-trained on a split QA-paired set. While the official codebase provided the script for pretraining, it did not include the exact base model (weights) they used. We executed the official script to generate the pre-trained base models and observed that meta-learning performance is sensitive to the choice of base models. For a fair comparison, we reported both the results from [66] (where CaMeLS [25] presented performance improvements over baselines using bar plots without specific metric values) and the results with our best custom pre-trained base models. Following the two-phase training paradigm introduced in Section 3.2, we performed training of $(FG)^2U$ on RGU (DistilGPT2, unrolled depth 6) results. The hyperparameters we used for $(FG)^2U$ are summarized in Appendix F.2, while all remaining hyperparameters were kept the same as in [25]. All experiments are conducted on one NVIDIA A100 GPU (80G).

| base model | DistilGPT2 | GPT2 |
|---|---|---|
| Unrolled Depth | 24/48 | 24/48 |
| # of Random Directions | 12 | 8 |
| Gradient Accumulate | 32 | 32 |
| Outer Optimizer | Adam | Adam |
| Outer Step Size | 2.5E-6 | 2.5E-6 |

Table F.2: $(FG)^2U$ hyperparameters for CaMeLS experiments.

## F.3  Data-driven Discovery of Partial Differential Equations (PDEs)

The hyperparameters we used for this experiment are summarized in Appendix F.3. All experiments are conducted on NVIDIA-L40S (40G). The structure for PINN is a depth-9 and width-20 MLP with tanh activations.

# G  Additional Experimental Results

## G.1  Meta Learning Online Adaptation of Language Models

Ablation results on the unrolled depth and the base model are summarized in Table G.1 and Table G.2.

| PDEs
Inner Solvers | Burgers | | AllenCahn | | KdV | |
|---|---|---|---|---|---|---|
| | PINN | Numerical | PINN | Numerical | PINN | Numerical |
| Directional Grad. Calculation | FAD | ZO | FAD | ZO | FAD | ZO |
| # Random Directions | 1 | 1 | 1 | 1 | 1 | 1 |
| Outer Steps | 5000 | 5000 | 5000 | 5000 | 5000 | 5000 |
| Unrolling Depth | 1000 | - | 1000 | - | 1000 | - |
| Grid Size for Numerical Method | - | 256×512 | - | 256×512 | - | 256×512 |
| Range of initial $\nu$ | (0, 1e1] | (0, 1e1] | (0, 1e-1] | (0, 1e-1] | (0, 1e-2] | (0, 1e-2] |
| Outer Optimizer | Adam | Adam | Adam | Adam | Adam | Adam |
| Outer Step Size | 1e-2 | 1e-2 | 1e-2 | 1e-2 | 1e-3 | 1e-3 |
| Inner Optimizer | SGD | - | SGD | - | SGD | - |
| Inner Batch Size | 5000 | - | 5000 | - | 5000 | - |
| Inner Step Size | 1e-3 | - | 1e-3 | - | 1e-3 | - |
| $\mu$ for Finite Difference | - | 1e-4 | - | 1e-4 | - | 1e-4 |

Table F.3: $(FG)^2U$ hyperparameters for discovery of PDEs experiments. $\nu$ denotes the unknown PDE parameters.

| Model (# params) | Method | Unrolled
Steps (#) | DistilGPT2 | | GPT2 | |
|---|---|---|---|---|---|---|
| | | | EM (↑) | F1 (↑) | EM (↑) | F1 (↑) |
| DistilGPT2 (82M) | RGU (impl.) | 6 | 2.04 | 5.53 | OOM | |
| | $(FG)^2U$ (ours) | 24 | 2.10 | 5.59 | **2.22** | **6.37** |
| | | 48 | 2.10 | 6.25 | 2.16 | 6.32 |
| GPT2-Large (774M) | RGU (impl.) | 6 | 7.02 | 12.19 | OOM | |
| | $(FG)^2U$ (ours) | 24 | 6.91 | 12.12 | 7.21 | **12.50** |
| | | 48 | 7.03 | 12.31 | **7.27** | 12.45 |
| GPT2-XL (1.5B) | RGU (impl.) | 6 | 7.93 | 12.94 | OOM | |
| | $(FG)^2U$ (ours) | 24 | 8.34 | 13.46 | **8.89** | **14.42** |
| | | 48 | 8.23 | 13.70 | 8.65 | 13.91 |

Table G.1: **StreamingQA**: Ablation results on the unrolled depth and the base model.

| Model (# params) | Method | Unrolled
Steps (#) | DistilGPT2 | | GPT2 | |
|---|---|---|---|---|---|---|
| | | | EM (↑) | F1 (↑) | EM (↑) | F1 (↑) |
| DistilGPT2 (82M) | RGU (impl.) | 6 | 1.52 | 3.16 | OOM | |
| | $(FG)^2U$ (ours) | 24 | 1.72 | 3.49 | 1.72 | **3.50** |
| | | 48 | **1.75** | 3.47 | 1.73 | 3.49 |
| GPT2-Large (774M) | RGU (impl.) | 6 | 4.86 | 8.57 | OOM | |
| | $(FG)^2U$ (ours) | 24 | 5.49 | 8.88 | **5.56** | **8.99** |
| | | 48 | 5.45 | 8.90 | 5.32 | 8.97 |
| GPT2-XL (1.5B) | RGU (impl.) | 6 | 6.71 | 9.65 | OOM | |
| | $(FG)^2U$ (ours) | 24 | 7.00 | 10.13 | 7.27 | 10.33 |
| | | 48 | **7.37** | **10.37** | 7.25 | 10.32 |

Table G.2: **SQuAD**: Ablation results on the unrolled depth and the base model.

