# OpenReview forum: "Memory-Efficient Gradient Unrolling for Large-Scale Bi-level Optimization"
_NeurIPS.cc/2024/Conference — NeurIPS 2024 poster_

### Official Review · Reviewer_J9FE · 2024-07-10

**Soundness:** 2
**Presentation:** 3
**Contribution:** 2
**Rating:** 5
**Confidence:** 4

**Summary:**

This paper studies scalable bi-level optimization problems and points out the limitations of most traditional methods. Among all, to mitigate the high memory cost issue of the GU method, it proposes (FG)2U such that the space consumption is reduced to $\mathcal{O}(M)$ from $\mathcal{O}(MN)$. Convergence analysis is provided and extensive experiments show the benefit of the algorithm.

**Strengths:**

1.	The presentation is great and I feel relaxed to read this paper.
2.	The experiment is extensive including both small and large scale settings.

**Weaknesses:**

1.	The memory cost is reduced but the computational cost seems to increase significantly. The computational cost takes the order of $\mathcal{O}(KTN)$ since $b=\mathcal{O}(N)$. This feels unrealistic as well in the large-scale application when $N$ is large.
2.	Based on the previous question, I checked the choice of $b$ in the experiment, which is small. How do authors tune this parameter? Is there any ablation study on it?
3.	In terms of the discussion in Appendix B.2, IF methods also require $\mathcal{O}(M)$ space using some approximation tricks. There indeed exist approximation errors in IF methods. However, normally, the approximation errors can be controlled to be very small based on the hyperparameter selections. Thus, IF methods seem suitable for scalable bi-level optimization problems to some extent. Do authors have time to include IF methods in Table 2?
4.	The algorithm is deterministic without consideration of the batch data. Can authors provide some insights in stochastic settings? For example, can the unbiasedness of the hypergradient estimator still be satisfied, or is there any requirement for the batch size?
5.	Can authors provide real space consumption in the comparison (Table 1, Table 2)?

**Questions:**

See weakness part.

**Limitations:**

See weakness part.

---

> ### Author Rebuttal · Authors · 2024-08-07
>
> We thank the reviewer for the overall positive evaluation and will try to address the concerns raised point by point.
>
> **Regarding the complexity**
>
> We direct the reviewer to the general response for computational cost analysis and discussions regarding the computation in practice, including the empirical success of FG/ZO in large-scale applications with constant batch size, tricks to reduce the variance, how the parallelizable nature of $\text{(FG)}^2$U fits modern AI computation, and the more cost-effective two-phase methodology in practice.
>
> **Regarding the strategy to choose $b$**
>
> We follow the methodology [2][3] developed for FG/ZO in large-scale applications to choose $b=\mathcal{O}(1)$ rather than $\mathcal{O}(N)$.
> We select the largest possible $b$ that does not exceed the GPU memory limit to fully utilize the GPU. See Tables C and D in the attached PDF in the general response for the exact memory consumption.
> Notice that we additionally perform gradient accumulation through iterative/multi-GPU parallel computation, which will also influence the variance.
> The number of accumulation steps is tuned in our initial attempts according to the wall-clock efficiency and stability.
>
> **Regarding the IF methods**
>
> Although the memory efficiency of IF methods makes them suitable for large-scale BO to some extent, there are several inherent limitations associated with the approximation of these methods. As discussed in Appendix B.2, the approximation bias due to unsatisfied KKT conditions cannot be controlled by hyperparameter tuning. Additionally, the challenge of stochastic approximation in IHVP is highlighted in [1]. Specifically, while the Hessian $H$ is approximated via mini-batch sampling, it holds that $H^{-1} = (E[H])^{-1} \neq E[H^{-1}]$. This corresponding bias also cannot be mitigated through hyperparameter tuning.
>
> In addition, following the reviewer's suggestion, we conducted additional experiments to complement the findings in Table 2, focusing on the comparison between IF methods and other approaches.
>
> We considered two IF methods: Hessian-free and Neumann.
> The Hessian-free method approximates the inverse Hessian as an identity matrix scaled by a scalar. The Neumann method approximates the inverse Hessian vector product by utilizing the Neumann Series (see Equation 25, noting that the exponent $k$ is missing for $(I - \alpha H)$).
>
> Due to the limited time available during the rebuttal period, we focused solely on the StreamingQA dataset. We fixed the base model as GPT2 and the unrolled depth at 48.
>
> For the Hessian-free method, the scalar can be merged into the learning rate since the explicit gradient defined in Equation 3 is zero in this case. We selected the learning rate from $[1E-4, 5E-5, 1E-5, 5E-6, 2.5E-6]$, with $5E-6$ yielding the best validation loss.
> For the Neumann method, we tuned the hyperparameters $\alpha$ and $K$ in Equation 25. We conducted a grid search with $(\alpha, K) \in [1, 0.1, 0.01, 0.001] \times [10, 20, 40]$ and a learning rate of $2.5E-6$. The optimal combination was $\alpha = 0.01$ and $K = 40$. No further performance improvement was observed by tuning the learning rate or increasing $K$ to 80.
>
> Please refer to Table A in the attached PDF in the general response for results. We observe the following:
>
> (1) Both IF methods yield improvements on RGU (with an unrolled depth of 6).
>
> (2) By carefully tuning the hyperparameters, the Neumann method achieves further improvements over the Hessian-Free method, albeit with additional computational cost.
>
> (3) Despite the careful hyperparameter tuning for Neumann, its performance remains inferior to $\text{(FG)}^2$U.
>
>
> **Regarding the stochastic $\text{(FG)}^2$U**
>
> In the stochastic context, the bilevel optimization problem is formulated as:
>
> $$\min_{\phi\in \Phi}\  h(\phi):=f(\theta_T(\phi), \phi)=E_{\xi}[F(\theta_T(\phi), \phi; \xi)]$$
>
> $$where\ \ \ \theta_0(\phi) = \Omega_0(\phi), \theta_t(\phi) = \Omega_t(\theta_{t-1}(\phi), \phi; \zeta)\in\Theta, t = 1,...,T, $$
>
> where $\xi$, $\zeta$ are random variables.
>
> Assume that the sampled dataset for the meta-objective and the lower-level objective are respectively $D_{F}$ and $D_{G}$, related to random variables $\xi$ and $\zeta$. We have the hypergradient under stochastic context:
>
>  $$\nabla_\phi ' h(\phi) = \frac{\partial F(\theta_T(\phi;D_G), \phi;D_F)}{\partial \theta_T} \frac{d\theta_T(\phi,D_G)}{d\phi} + {\frac{\partial F(\theta_T(\phi;D_G), \phi;D_F)}{\partial \phi}}.$$
> With Forward Gradient, the estimation of hypergradient would be $\nabla_\phi ' h(\phi) vv^{\top}$. Subsequently, from the independence between the forward gradient vectors $v$ and the batch sampling, we have
> $$E[\nabla_\phi ' h(\phi)] = E_{\xi,\zeta}[E_{v}[\nabla_\phi ' h(\phi)vv^{\top}]]= E_{\xi,\zeta}[\nabla_\phi ' h(\phi)]$$
>
> $$=E_{\xi,\zeta}\left[\frac{\partial F(\theta_T(\phi;D_G), \phi;D_F)}{\partial \theta_T} \frac{d\theta_T(\phi,D_G)}{d\phi} + {\frac{\partial F(\theta_T(\phi;D_G), \phi;D_F)}{\partial \phi}}\right]$$
>
> $$= \frac{\partial f(\theta_T(\phi), \phi)}{\partial \theta_T} \frac{d\theta_T(\phi)}{d\phi}+ \frac{\partial f(\theta_T(\phi), \phi)}{\partial \phi}=\nabla_\phi h(\phi),$$
> which gives the unbiasedness of $\text{(FG)}^2$U for stochastic bilevel optimization problems.
>
> **Regarding the real space consumption**
>
> We direct the reviewer to Tables B, C, and D in the attached PDF in the general response.
>
> [1] Making Scalable Meta Learning Practical, https://arxiv.org/abs/2310.05674
>
> [2] Fine-Tuning Language Models with Just Forward Passes, https://arxiv.org/abs/2305.17333
>
> [3] Revisiting Zeroth-Order Optimization for Memory-Efficient LLM Fine-Tuning: A Benchmark, https://arxiv.org/abs/2402.11592

---

> > ### Comment · Reviewer_J9FE · 2024-08-11
> > **Thanks for the response**
> >
> > I thank the authors for their response. Some of my concerns have been addressed. However, I am still slightly concerned about the theoretical guarantee on the problem dimension. There are also some works on bilevel optimization using zeroth-order types of methods. Maybe the authors would like to include them and provide a comparison. I keep my current score.
> >
> > Best,
> > Reviewer

---

> > > ### Author Response · Authors · 2024-08-12
> > > **Thanks again for your valuable feedback**
> > >
> > > Thank you for your response. We regret to hear that some concerns remain.
> > >
> > > Regarding the convergence guarantee dependent on the problem dimension, we acknowledge that further improvement of the theoretical results would require assumptions that may be impractical. In the field of backpropagation-free optimization (FG/ZO), the gap between dimension-dependent theoretical results and the significantly positive empirical outcomes in large-scale cases remains an open question. We hope that future research will help bridge this gap.
> > >
> > > Regarding BO + ZO, we followed the reviewer’s suggestions and identified several related works [1][2][3]. Given the limited time remaining in the discussion period, we are unable to conduct a comprehensive empirical study of these methods. However, we offer some preliminary comments here and will seriously consider incorporating comparisons in our revised paper.
> > >
> > > 1. [1][3] propose utilizing zeroth-order Hessian/Jacobian approximations for IF-based methods, whereas our work focuses on GU-based methods. It is important to note that zeroth-order approximation cannot eliminate the inherent bias introduced by IF-based methods, and the theoretical guarantees provided in these works are also dimension-dependent.
> > >
> > > 2. [2] employs zeroth-order optimization in a GU manner. However, [2] is limited to Neural Architecture Search rather than universal BO. Additionally, the inner problem considered in [2] is differentiable (white box), while our exploration of $($FG$)^2$U-ZO is in the more challenging black-box setting.
> > >
> > > We would like to once again extend our sincere thanks for your valuable feedback. We remain open to answering any further questions during the remaining time of the discussion period.
> > >
> > > $ $
> > >
> > > [1] On the Convergence Theory for Hessian-Free Bilevel Algorithms, https://arxiv.org/abs/2110.07004
> > >
> > > [2] ZARTS: On Zero-order Optimization for Neural Architecture Search, https://arxiv.org/abs/2110.04743
> > >
> > > [3] Fully Zeroth-Order Bilevel Programming via Gaussian Smoothing, https://arxiv.org/pdf/2404.00158

---

> > > > ### Comment · Reviewer_J9FE · 2024-08-12
> > > >
> > > > Thank the authors for their further clarifications.
> > > >
> > > > Best,
> > > >
> > > > Reviewer

---

### Official Review · Reviewer_4Mx8 · 2024-07-10

**Soundness:** 2
**Presentation:** 3
**Contribution:** 2
**Rating:** 5
**Confidence:** 4

**Summary:**

The paper introduces a method called Forward Gradient Unrolling with Forward Gradient, abbreviated as (FG)²U, which is designed to address the large memory requirements of forward method in bi-level optimization in large-scale machine learning model

**Strengths:**

1. The method significantly reduces memory overhead compared to traditional gradient unrolling methods, making it suitable for large-scale applications.
2. Can be easily implemented within popular deep learning frameworks and adapted to various optimization problems.

**Weaknesses:**

1. The proposed method introduces additional computational complexity. Can the author give an analysis of the complexity?
2. The convergences analysis is for the problem (2) not the original problem (1).
3. More large scale datasets are needed.

**Questions:**

see weakness

---

> ### Author Rebuttal · Authors · 2024-08-07
>
> We thank the reviewer for the feedback. We will try to address the concerns point by point.
>
> **Regarding the computational complexity**
>
> We direct the reviewer to our general response. We place a computational cost analysis in A.1 and a more detailed discussion on practical computation in A.2.
>
> **Regarding the convergence of problem (1)**
>
> To extend the convergence of optimization problem (2) into (1), we need to assume that the lower-level objective function $g$ is strongly convex w.r.t. $\theta$ as commonly done by previous works [2][3]. From the strong convexity and first-order smoothness (Assumption 3.2) of $g$, we have 1) the zeroth and first-order smoothness of $\theta^*(\phi)$; 2) $\|\theta_T(\phi')-\theta^*(\phi')\|\to 0$ as $T\to +\infty$.
> Then the inequality
> $$\|\theta_T(\phi)-\theta_T(\phi')\|\leq \|\theta_T(\phi)-\theta^*(\phi)\|+\|\theta_T(\phi')-\theta^*(\phi')\|+\|\theta^*(\phi)-\theta^*(\phi')\|$$
> implies the the zeroth and first-order smoothness of $\theta_T(\phi)$. Following the same line of proof as presented in our paper, we derive the smoothness of $f(\theta^*(\phi),\phi)$ and $f(\theta_T(\phi),\phi)$, and subsequently the convergence of either problem (1) or (2).
>
> However, as discussed in lines 73 to 77, given that the scope of this paper is large-scale BO, the inner optimization typically involves deep neural networks.
> Therefore, the optimal parameters are not explicitly accessible and can only be estimated through iterative procedures.
> Most related works [1][2][3] are implicitly or explicitly solving (2) instead of (1).
>
> Additionally, it is important to acknowledge that achieving strong convexity is often unfeasible in practical applications.
> Consequently, in this paper, we aim to present a more practical convergence theory that proves the effectiveness of our method.
>
> **Regarding larger-scale datasets**
>
> As discussed in Appendix H, this work serves as the first attempt to apply FG/ZO in BO.
> While the scale of cases studied in this paper is relatively small compared to the most powerful generative models in the industry, it is comparable to recent large-scale BO works such as [1] and significantly larger than traditional BO works.
> We believe the empirical studies conducted in this paper are sufficient to demonstrate the potential of $\text{(FG)}^2$U in large-scale BO.
> We anticipate that the effectiveness of $\text{(FG)}^2$U will be further validated in larger scales.
>
> [1] Making Scalable Meta Learning Practical, https://arxiv.org/abs/2310.05674
>
> [2] Truncated Back-propagation for Bilevel Optimization, https://arxiv.org/abs/1810.10667
>
> [3] Bilevel Optimization: Convergence Analysis and Enhanced Design, https://arxiv.org/abs/2010.07962

---

> > ### Comment · Reviewer_4Mx8 · 2024-08-11
> > **Response to author**
> >
> > Thank you for the rebuttal, the authors have addressed all my concerns. I will increase my score.

---

> > > ### Author Response · Authors · 2024-08-12
> > > **Thanks again**
> > >
> > > We are pleased to hear that our response has addressed your concerns. We would like to extend our sincere thanks for your valuable feedback again.

---

### Official Review · Reviewer_cwe5 · 2024-07-11

**Soundness:** 3
**Presentation:** 4
**Contribution:** 3
**Rating:** 6
**Confidence:** 3

**Summary:**

This paper presents a novel gradient unrolling algorithm for bi-level optimization. The authors highlight that existing methods for calculating meta gradients in the literature are not memory efficient. They propose a sampling-based method, (FG)^2U, to estimate the meta gradient. This approach approximates the meta gradient by multiplying it with a random rank-1 matrix, thereby simplifying the Forward Gradient Unrolling (FGU) scheme. The paper includes discussions on sampling efficiency, convergence analysis, and numerical experiments for (FG)^2U.

**Strengths:**

- The proposed method (FG)^2U is simple and easy to implement.
- The writing is clear and easy to follow.
- The theoretical results clearly demonstrate the relationship between convergence, sampling batch size, and parameter space size.
- The three experiments, data condensation, LM fine-tuning, and PDE recovery, show the wide range of applications of bi-level optimization and the proposed method (FG)^2U.

**Weaknesses:**

A major concern is the theoretical dependence on the parameter dimension $N$. Theorem 3.4 indicates that the convergence rate depends linearly on N. It means that
- With a fixed batch size, convergence on large-scale applications would be slow (vanilla (FG)^2U without additional techniques).
- With an $O(N)$ batch size, convergence is satisfactory, but the memory cost grows as $O(MN)$, similar to FGU.
- While the authors mention that (FG)^2U allows parallelization to mitigate computational overhead, it seems that the calculations of FGU (4) also permit parallelization, correct? Specifically, $d \theta / d \phi$ can be computed in parallel as $d \theta[1] / d \phi, d \theta[2] / d \phi, d \theta[3] / d \phi, \cdots$

**Questions:**

Regarding the formula for data condensation (17) and related formulas in the appendix: should it be minimizing over ${\mathcal{D}_c}$ instead of minimizing over ${\mathcal{D}_o}$?

**Limitations:**

The paper appears to have no potential negative societal impact. The authors discussed the limitations in the Appendix H.

---

> ### Author Rebuttal · Authors · 2024-08-07
>
> We thank the reviewer for the overall positive evaluation and will try to address the concerns raised point by point.
>
> **Regarding the complexity**
>
> The reviewer's understanding of Theorem 3.4 is generally correct. We would like to highlight the following points:
>
> Firstly, the convergence rate in Theorem 3.4 is an upper bound. We direct the reviewer to our general response (A.2) for a more detailed discussion on the choice of $b$ and the computation in practice, covering the empirical success of FG/ZO in large-scale applications with constant batch sizes, techniques to reduce variance, the parallelizable nature of $\text{(FG)}^2$U which aligns well with modern AI computation, and the more cost-effective two-phase methodology in practice.
>
> Secondly, the $\mathcal{O}(bM)$ memory cost with batch size $b$ can be reduced to $\mathcal{O}(M)$ through serial/parallel accumulation. In other words, we do not have to compute the full batch on a single GPU simultaneously. In practice, the memory cost of $\text{(FG)}^2\text{U}$ can be more manageable than FGU and RGU by choosing $b$ according to the hardware. It is important to note that both serial and parallel accumulation are not trivial for FGU and RGU.
>
> Thirdly, the parallelization method for FGU (Plan A) proposed by the reviewer can be expressed as $d\theta / d\phi = Z = I Z = \sum_i^M \text{diag}(e_i) Z = \sum_i^M e_ie_i^T Z$.
> As mentioned in Line 122, a special choice of $v$ is the normalized orthogonal coordinates (Plan B), which leads to $Z vv^T = N Z e_j e^T_j$, where $j$ is a random index from $Unif([0,\ldots,N])$.
>
> The differences are:
>
> (1) Plan A maintains the rows of $Z$, while Plan B maintains the columns.
>
> (2) Plan B utilizes randomization, while Plan A does not.
>
> We argue that both choices of Plan B are better:
>
> (1) Considering the update rule in Equation 4, row-wise updates require communication among threads, whereas column-wise updates do not.
>
> (2) Plan A requires exactly $M$ threads, whereas Plan B provides smooth trade-offs via randomization.
>
> **Regarding the formula for data condensation**
>
> Yes, the reviewer is correct. We will fix the typo in the revised manuscript.

---

> > ### Comment · Reviewer_cwe5 · 2024-08-12
> > **Response to authors**
> >
> > I greatly appreciate the efforts made by the authors during the rebuttal phase, including their responses to my questions and the inclusion of additional experimental results.
> >
> > However, my primary theoretical concern, which I raised in the initial review, remains only partially addressed.
> >
> > The statement "Plan A requires exactly $M$ threads" may not be entirely accurate. It is possible to adaptively allocate the $M$ calculation tasks based on the constraints of the hardware. For instance, if $M=5$, one could opt for 3 threads and distribute the tasks as $[1,1,3]$, rather than being limited to $[1,1,1,1,1]$. This flexibility also casts doubt on the assertion that "the memory cost of $\text{(FG)}^2\text{U}$ can be more manageable than FGU and RGU," as one could potentially implement simple variants of FGU or RGU.
> >
> > Of course, as the authors have noted, these analyses are merely upper bounds. The actual performance in the experiments presented by the authors appears promising. Therefore, I will keep my positive score.

---

> > > ### Author Response · Authors · 2024-08-12
> > > **Thanks again for your valuable feedback**
> > >
> > > Thank you for your response and for the recognition of our work.
> > >
> > > Regarding the convergence guarantee dependent on the problem dimension, we acknowledge that further improvement of the theoretical results would require assumptions that may be impractical. In the field of backpropagation-free optimization (FG/ZO), the gap between dimension-dependent theoretical results and the significantly positive empirical outcomes in large-scale cases remains an open question. We hope that future research will help bridge this gap.
> > >
> > > Regarding the parallelization plan, the essential information we intended to convey in our rebuttal response is that "a special parallelization plan is covered by $($FG$)^2$U as a universal framework." We acknowledge that we may have overstated the advantages of $($FG$)^2$U over FGU and RGU in terms of memory management, as pointed out by the reviewer. We will exercise caution with related statements in our revised paper.
> > >
> > > We would like to once again extend our sincere thanks for your valuable feedback. We remain open to answering any further questions during the remaining time of the discussion period.

---

### Author Rebuttal · Authors · 2024-08-07

## (A) General Response

We thank all reviewers for their constructive feedback.
In this response, we will address some common concerns raised by the reviewers.
We will revise the manuscript covering the following discussions.

All reviewers raised concerns about the computation.
More specifically, Reviewer 4Mx8 specifically requested an analysis of the complexity. Reviewer cwe5 and Reviewer J9FE raised concerns about the dimension-dependent convergence rate of forward gradient (FG).
We will begin with a theoretical computation cost analysis of GU-based methods, then move to A.2 for discussions on practical computation.

### (A.1) Theoretical Computational Cost

We follow existing works [4][5] in treating the transitions $\Omega_t: \theta_{t-1} \rightarrow \theta_{t}$ as indivisible computational units and conduct the computational cost analysis by focusing on Jacobian (jac), Jacobian-vector product (jvp), Hessian-vector product (hvp), vector-Jacobian product (vjp), vector-Hessian product (vhp), and Hessian-Jacobian product (hjp) operations around $\Omega$.

The gradient computation costs are as follows:

(i) FGU involves $T$ hjp and $T$ jac operations (Equation 4).

(ii) RGU involves $T$ vjp and $T$ vhp operations (Equation 6).

(iii) $\text{(FG)}^2$U involves $bT$ hvp and $bT$ jvp operations (Equation 9).

Overall, the computational complexities of FGU, RGU, and $\text{(FG)}^2$U are $\mathcal{O}(MNT)$, $\mathcal{O}(MT)$, and $\mathcal{O}(bMT)$, respectively.

Here are some discussions:

(a) $b=\mathcal{O}(N)$ or $\frac{N}{b}$ times updates are required to achieve convergence for $\text{(FG)}^2$U. The total computation will be $\mathcal{O}(MNT)$.

(b) Notice (a) is about the theoretical upper bound, we will further discuss the computation cost in practice in (A.2).

(c) RGU is the most computationally efficient. However, as discussed in lines 105-111, memory issues impede RGU in large-scale scenarios. $\text{(FG)}^2$U improves memory performance at the cost of increased computation.

$ $
### (A.2) Practical Computational Cost

**Firstly, the dimension-dependent convergence rate is an upper bound and the scalability has proven acceptable in practice**.
As evidence, zeroth-order optimization (ZO), which can be regarded as a finite difference approximation of FG, has been used as a standard technique for fine-tuning large language models (LLMs) [1][2].
The size of the LLMs studied is up to 66B, and the fine-tuning performance is competitive with full-parameter fine-tuning, with $b=\mathcal{O}(1)$ ($b=1$ in [1] and $b=16$ in [2]).

**Secondly, the variance of the FG/ZO can be reduced by various tricks.**
[2] explored ZO + gradient pruning, which reduced the effective number of dimensions based on the lottery hyperthesis.
[3] propose ZO + SVRG to reduce the variance.
We didn't explore these tricks in our initial attempt to apply FG/ZO in BO, but we believe these are promising directions for our future works to scale up $\text{(FG)}^2$U and $\text{(FG)}^2$U-ZO, as we have discussed in Appendix H.

**Further, $\text{(FG)}^2$U is suitable for modern AI computation due to its parallelizable nature.**
With the practice of scaling laws and the development of AI infrastructure, the computational cost concerns of large-scale models are mitigated when efficient parallelization is available.
A vivid example is Transformer, whose quadratic complexity in sequence length is less favorable compared to RNNs, yet it has achieved impressive empirical success.
The key to scaling up the Transformer lies in the parallelizable nature of attention.

Similarly, the inherently parallelizable and hardware-friendly nature of $\text{(FG)}^2$U enables it to leverage large-scale distributed computing resources (as mentioned in lines 146-148) within popular deep learning frameworks (as discussed in lines 196-204), with minimal engineering effort.

**Furthermore, the more cost-effective two-phase methodology can be utilized to reduce the overall computational expense.**
Considering the computational cost, we do not recommend using $\text{(FG)}^2$U from scratch, as discussed in Sec. 3.2.
Instead, we advocate for a two-phase methodology: initially, employing efficient yet less accurate gradient approximation methods, such as TRGU or Hessian-Free, and subsequently using $\text{(FG)}^2$U for more accurate, albeit less efficient, gradient approximation to further enhance performance.

Finally, we want to emphasize the scope of this work: $\text{(FG)}^2$U is intended to complement, rather than overturn, the existing methodology of BO.
The trade-off between computation and performance is a perpetual theme in computer science.
Within the BO community, previous works have tended to sacrifice performance for reduced computational costs.
We believe $\text{(FG)}^2$U will bring new insights to the BO community, prompting a reconsideration of this trade-off, especially in the current era of rapidly developing AI infrastructure and scaling law methodologies.

[1] Fine-Tuning Language Models with Just Forward Passes, https://arxiv.org/abs/2305.17333

[2] Revisiting Zeroth-Order Optimization for Memory-Efficient LLM Fine-Tuning: A Benchmark, https://arxiv.org/abs/2402.11592

[3] Variance-reduced Zeroth-Order Methods for Fine-Tuning Language Models, https://arxiv.org/abs/2404.08080v1

[4] Truncated Back-propagation for Bilevel Optimization, https://arxiv.org/abs/1810.10667

[5] Bilevel Optimization: Convergence Analysis and Enhanced Design, https://arxiv.org/abs/2010.07962

## (B) Additional Experiments

Please refer the the attached PDF for additional results, including comparisons to additional baselines and real memory consumptions.

---

### Decision · Program_Chairs · 2024-09-25

**Decision:**

Accept (poster)

**Comment:**

This paper introduces a Forward Gradient Unrolling with Forward Gradient, abbreviated as (FG)2U, achieving an unbiased stochastic approximation of the meta gradient. All the reviewers agree to accept this paper based on reasons of (1) simple and easy to implement, (2) well linked theoretical contribution, (3) comprehensive experiments, etc. I mostly agree.

Some concerns are also raised: (1) Theorem 3.4 indicates that the convergence rate depends linearly on N and states the various useful implications about it; (2) the computational cost takes the order of O(N) and it is suggested to add discussion on the zero-order BO.

Thus, I recommend an accept subject to addressing the mentioned issues in the final version